# A potent Henipavirus cross-neutralizing antibody reveals a dynamic fusion-triggering pattern of the G-tetramer

Pengfei Fan [1,9] ✉, Mengmeng Sun[2,9], Xinghai Zhang[3,9], Huajun Zhang [3], Yujiao Liu[1], Yanfeng Yao [3], Ming Li[2], Ting Fang[1], Bingjie Sun[1], Zhengshan Chen[1], Xiangyang Chi[1], Li Chen [3,4], Cheng Peng[3], Zhen Chen[3], Guanying Zhang[1], Yi Ren[1], Zixuan Liu[1], Yaohui Li[1], Jianmin Li[1], Entao Li[2], Wuxiang Guan[3], Shanshan Li [2,5,6], Rui Gong [3] ✉, Kaiming Zhang [2,5,6] ✉, Changming Yu [1] ✉ & Sandra Chiu [2,7,8] ✉

The Hendra and Nipah viruses (HNVs) are highly pathogenic pathogens without approved interventions for human use. In addition, the interaction pattern between the attachment (G) and fusion (F) glycoproteins required for virus entry remains unclear. Here, we isolate a panel of Macaca-derived G-specific antibodies that cross-neutralize HNVs via multiple mechanisms. The most potent antibody, 1E5, confers adequate protection against the Nipah virus challenge in female hamsters. Crystallography demonstrates that 1E5 has a highly similar binding pattern to the receptor. In cryo-electron microscopy studies, the tendency of 1E5 to bind to the upper or lower heads results in two distinct quaternary structures of G. Furthermore, we identify the extended outer loop β1S2-β1S3 of G and two pockets on the apical region of fusion (F) glycoprotein as the essential sites for G-F interactions. This work highlights promising drug candidates against HNVs and contributes deeper insights into the viruses.

The Nipah (NiV) and Hendra (HeV) viruses are nonsegmented single-stranded RNA viruses belonging to the genus *Henipavirus* of the *Paramyxoviridae* family. According to standardized genotyping methods, NiV can be divided into two major strains, NiV Malaysia (NiV_{MY}) and NiV Bangladesh (NiV_{BD})[1]. Both Hendra and Nipah viruses (HNVs) can cause acute and severe respiratory illnesses and encephalitis in humans, with fatality rates ranging from 40 to 100%[2]. Unlike other members of the family, HNVs have a wide susceptibility range involving at least six species of Chiroptera and ten species of five other mammalian orders (Artiodactyla, Perissodactyla, Carnivora, Primates, and Rodentia)[3–7]. More than 20 countries are threatened by HNVs, based on outbreaks, serological evidence, molecular detection, and the home range of *Pteropus* bats, leaving approximately 2 billion people at risk of virus spillovers[8]. Since being discovered in the 1990s,

[1]Laboratory of Advanced Biotechnology, Institute of Biotechnology, Beijing, China. [2]Division of Life Sciences and Medicine, University of Science and Technology of China, Hefei, China. [3]State Key Laboratory of Virology, Wuhan Institute of Virology, Center for Biosafety Mega-Science, Chinese Academy of Sciences, Wuhan, China. [4]University of Chinese Academy of Sciences, Beijing, China. [5]Center for Advanced Interdisciplinary Science and Biomedicine of IHM, MOE Key Laboratory for Cellular Dynamics, Division of Life Sciences and Medicine, University of Science and Technology of China, Hefei, China. [6]Department of Urology, The First Affiliated Hospital of USTC, Division of Life Sciences and Medicine, University of Science and Technology of China, Hefei, China. [7]Department of Laboratory Medicine, The First Affiliated Hospital of USTC, Division of Life Sciences and Medicine, University of Science and Technology of China, Hefei, China. [8]Key Laboratory of Anhui Province for Emerging and Reemerging Infectious Diseases, Hefei 230027 Anhui, China. [9]These authors contributed equally: Pengfei Fan, Mengmeng Sun, Xinghai Zhang. ✉e-mail: fanpengfei93@163.com; gongr@wh.iov.cn; kmzhang@ustc.edu.cn; yuchangming@126.com; qiux@ustc.edu.cn

HNVs have caused several outbreaks in humans and livestock in Australia, Bangladesh, Malaysia, and Singapore[9,10]. In a recent Nipah virus outbreak in Kerala, India, two of the six infected people died[11]. Although high lethality limits the opportunity for the virus to spread rapidly through populations, the circulation among multiple host species and its presence in lung and nasopharyngeal secretions during acute infection could increase its contagiousness[9]. However, no approved vaccines or therapeutic options are available for human use against HNV infections[12].

During viral entry, the attachment (G) glycoprotein facilitates the attachment of the virus to the host cell by interacting with the receptors ephrin-B2 (EB2) and ephrin-B3 (EB3) and triggers membrane fusion mediated by the fusion (F) glycoprotein[13,14]. Because of their critical role in viral invasion, G and F are major targets for developing therapeutic monoclonal antibodies (mAbs). Several neutralizing or cross-neutralizing antibodies targeting G or F that exhibit ideal animal protection have been isolated[4,6,15–21]. The efficacy, safety, and tolerability of m102.4 in compassionate use and phase I clinical trials further confirmed the benefits and potential of antibody therapies for HNV infections[12,22]. However, loss of neutralizing efficacy of antibodies due to viral variation has been observed[23,24]. Similarly, neutralization-escape mutants of the mAbs m102.4[22], nAH1.3[25], and h5B3.1[11] were isolated in vitro or in vivo. In addition, multiple new species or strains of henipavirus have been identified in recent years[5,26–29], and the nature of host-adapted evolution[30,31] and error-prone replication[6,32] may further enrich virus populations. Moreover, despite reports on the structures of EB2/3[33,34], G[35], and F[36], the dynamic interaction patterns between the G-tetramer and receptor, receptor binding domain (RBD)-targeting antibodies, or F proteins have not been determined.

In this work, we characterize a group of G-specific HNVs cross-neutralizing antibodies with diverse epitopes and mechanisms, expanding the currently limited drug-candidate library for emergencies. In crystallography, the most potent antibody, 1E5, exhibits a highly similar binding pattern to the receptor. The cryo-electron microscopy (cryo-EM) structures demonstrate that the upper binding of the receptor-mimicking 1E5 can disrupt the stability of the G-tetramer, which may be the molecular basis for G-mediated F-triggering signal transfer. We further determine the potential interaction sites between G and F to provide valuable insights into the virus invasion.

## Results

### Screening of antibodies from an immunized rhesus monkey

To obtain cross-reactive antibodies against HNVs, we sequentially immunized a female *Macaca mulatta* (rhesus macaque) with rAd5-NiV$_{BD}$ (adenovirus type 5 encoding full-length NiV$_{BD}$ G)[37] and the recombinant ectodomain of NiV$_{BD}$ and HeV G (Fig. 1a). Serum was collected at various time points to detect binding titres for HNV G and neutralizing titres against rHIV-HNVs (human immunodeficiency virus backbone pseudoviruses bearing the HNV G and F glycoproteins). Immunization with a single dose of rAd5-NiV$_{BD}$ elicited limited HNV G-specific antibody levels and failed to induce the production of rHIV-HeV neutralizing antibodies (Fig. 1b, c). Because NiV$_{BD}$ and NiV$_{MY}$ G share highly similar amino acid contents (approximately 96%), the trend in the serum antibody levels against the two NiV strains was similar after each immunization. In comparison, the antibody titre against HeV was relatively low after the first two immunizations, and this difference narrowed after the final HeV G inoculation. We also tested the ability of the serum to block G protein binding to the receptor using a previously described method (Fig. 1d)[38]. Seven days after NiV$_{BD}$ G stimulation, a considerable number of antibodies targeting the NiV G receptor binding domain were present, whereas these antibodies did not increase after the HeV G booster. The serum did not block the binding of EB2 to HeV G throughout the study period, although the binding and neutralization titres increased after HeV G vaccination (Fig. 1b–d).

We sorted memory B cells labeled CD3$^-$CD19$^+$CD27$^+$IgG$^+$NiV$_{BD}$ G$^+$ from peripheral blood mononuclear cells (PBMCs) collected on day 77 (Fig. 1e) and recovered 254 natural pairs of variable genes of the immunoglobulin heavy/light chain through single-cell polymerase chain reaction (PCR)[39]. These sequences were assembled into linear cassettes containing a human IgG$_1$ constant region for rapid transient expression in HEK293T cells[40]. We identified 88 NiV$_{BD}$ G-positive clones (Fig. 1f), 85 of which were encoded by unique genes (Fig. 1g, Supplementary Fig. 1a). The gene family distribution of the V(D)J sequences was consistent with that of previous studies on rhesus macaques[39,41]. IGHV4 (58.8%, 50/85), IGHD3 (35.3%, 30/85), and IGHJ4 (43.5%, 37/85) were the dominant VDJ families for V$_H$ (Supplementary Fig. 1b–d). IGKV1 (52.3%, 34/65) was the dominant V family for V$_\kappa$, and four J families, IGKJ1, IGKJ2, IGKJ3, and IGKJ4, with close proportions, constituted the J gene usage (Supplementary Fig. 1e). In contrast, V$_\lambda$, which was only one-third of the number of V$_\kappa$ genes, showed greater diversity, using eight V families and five J gene families, and had no obvious prevailing VJ family (Supplementary Fig. 1e).

### Characterization of functional antibodies

We produced chimeric macaque-human antibodies in mammalian cells and evaluated their neutralization capacity. Eight mAbs completely inhibited rHIV-NiV$_{BD}$ at a concentration of 1 µg ml$^{-1}$ (Supplementary Fig. 1f). In this study, we profiled the biological properties of eight mAbs in detail. All the mAbs showed high identities with germline genes, ranging from 90.3 to 97.2% for V$_H$ and from 89.3 to 98.7% for V$_L$ (Fig. 1h). To ascertain whether mAbs recognize different antigenic regions on the G surface, we performed a competition binding assay using streptavidin probes through biolayer interferometry (BLI) assays (Fig. 1i, Supplementary Fig. 2a). We segregated the eight mAbs into three major groups according to the competition: groups A (1A9 and 1F9), B (1B6 and 2E7), and C (2A4, 1D11, 1E5, and 2B8). The antibodies in group C competed strongly with m102.4, a broadly neutralizing antibody (bnAb) that binds to the RBD[42], suggesting that their epitopes were located at or spatially close to the receptor-binding domain.

To determine the binding breadth of the mAbs to recombinant HNV G proteins, we performed binding assays using recombinant HNV G proteins containing a full-length ectodomain (71–602 aa for NiV G and 71–604 aa for HeV G) or a recombinant NiV$_{BD}$ G head domain (G$_{HD}$, 176–602 aa) lacking the stalk and neck regions (Supplementary Fig. 2b). All the mAbs bound to NiV$_{BD}$ G at low concentrations, with a half-maximal effective concentration (EC$_{50}$) ≤ 10 ng ml$^{-1}$ (Fig. 2a–d, Table 1, Supplementary Fig. 3a). Seven of the eight mAbs bound to NiV$_{MY}$ G at comparably low concentrations, except for one antibody, 1A9, which was unable to bind. Five mAbs from groups B or C exhibited strong cross-reactivity with HeV G, with EC$_{50}$ values ranging from 2.8 to 33.2 ng ml$^{-1}$. No reduction in the binding ability of mAbs to monomeric G$_{HD}$ was observed, indicating that the antibodies did not target the stalk or neck regions.

To determine the capacity of the mAbs to recognize natural G-tetramers, we displayed full-length NiV$_{BD}$ G on the surface of 293T cells and incubated the cells with 5 µg ml$^{-1}$ antibodies for flow cytometric analysis. All the mAbs recognized the membrane-anchored natural G structure, and the difference in the proportion of positive cells, ranging from 58.8 to 83.8%, may be related to their epitope accessibility (Fig. 2e, Table 1, Supplementary Fig. 3b). Furthermore, we evaluated the kinetics of mAb binding to HNV G ectodomains by BLI using anti-human Fc probes. The specificity and binding ability of the antibodies were consistent with those determined by ELISA (Fig. 2a–d, Table 1, Supplementary Fig. 3a). All the antibodies showed rapid association (kon) and slow dissociation (koff) for their binding ligands (Supplementary Fig. 4). Determination of the equilibrium dissociation constant (K$_D$) revealed that these mAbs bound HNVs G with a nanomolar or subnanomolar avidity (Table 1, Supplementary

 

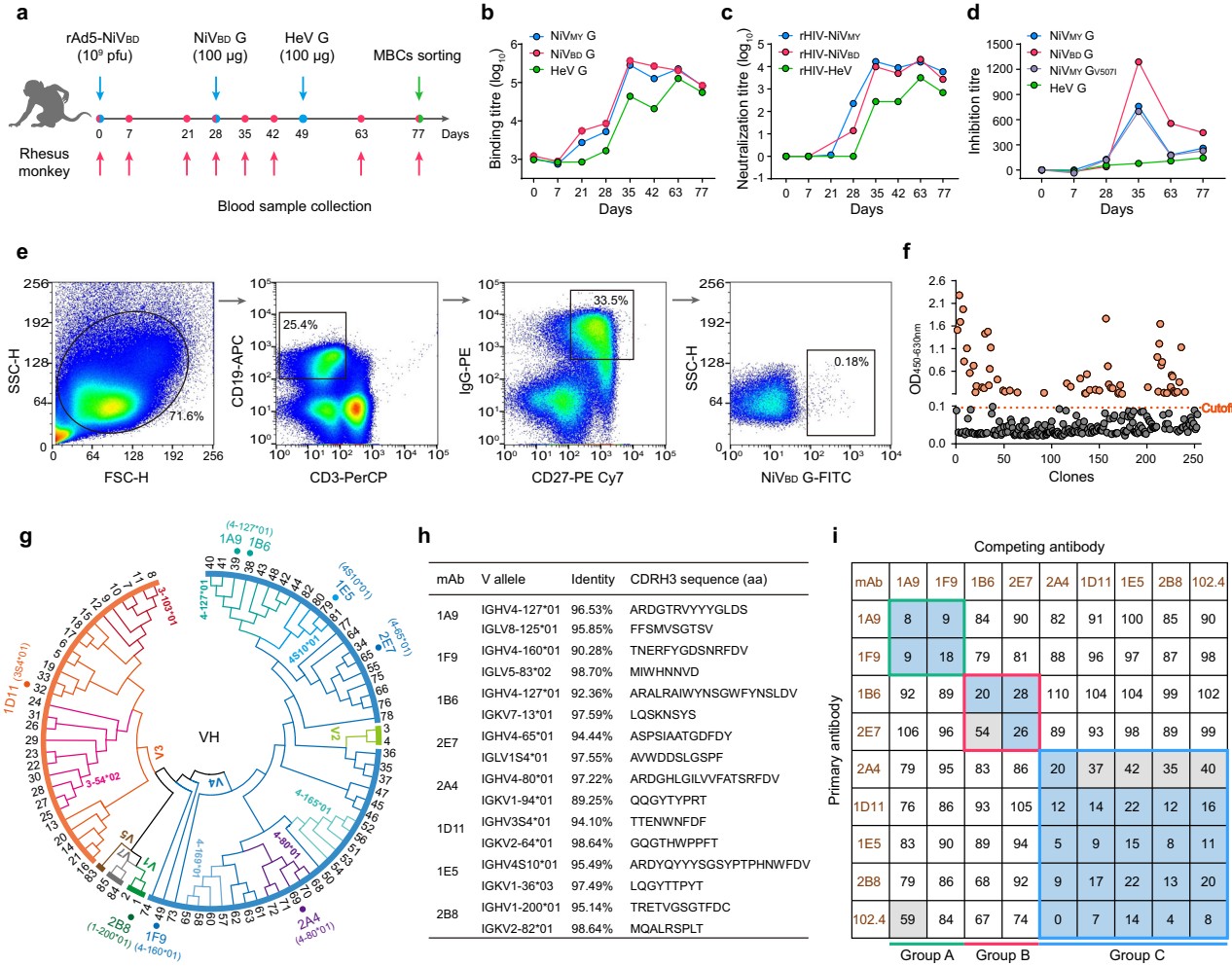

**Fig. 1 | NiV_BD G-specific antibodies were isolated from a vaccinated rhesus monkey. a** Timeline of immunization (blue arrows), serum collection (magenta arrows), and PBMC isolation (green arrow). **b** The binding titre of the serum at each time point to the G ectodomain of NiV_MY, NiV_BD, and HeV was tested by ELISA. Data are average values of two replicates. **c** The neutralizing titres of the sera against pseudotyped rHIV-NiV_MY, -NiV_BD, and -HeV. Each dot represents the mean reciprocal of serum dilution at half inhibition (two replicates for rHIV-NiV_MY and -HeV, and three for rHIV-NiV_BD), and the value was defined as 1 (10⁰) when half inhibition could not be achieved. **d** Capacity of sera to block EB2:HNV G binding in Luminex assays. Each dot represents the reciprocal of the serum dilution at half inhibition. **e** Sorting of single NiV_BD G specific memory B-cell by flow cytometry. The gating strategies and cell proportions are indicated. **f** Rapid identification of NiV_BD G-binding antibodies through ELISA using linear expression cassettes. The cutoff value was defined as 2.1 times the optical density of the control wells. **g** Phylogeny of variable genes of functional antibodies based on VDJ amino acid sequences (see also Supplementary Fig. 1a). Germline genes of neutralizing antibodies or with over four frequencies are specially marked. **h** Germline identity and CDRH3 sequence of eight neutralizing antibodies. **i** Competition-binding assay data determined by BLI (see also Supplementary Fig. 2a). Numbers in boxes indicate the percent binding of the second mAb in the presence of the first mAb compared to binding of the second mAb alone. The binding competition was determined based on the percent binding value: <33.3%, strong competition (blue boxes); 33.3–66.7%, intermediate competition (gray boxes); >66.7%, noncompetition (white boxes). Inferred competition-binding groups A to C are framed with green, magenta, and blue, respectively. Source data are provided as a Source Data file.

Table 1). Except for mAb 1D11, which dissociated after binding to HeV G, the other antibodies barely dissociated once bound to the analyte.

We performed dose−response inhibition assays using rHIV-HNVs to evaluate the broad potency of the eight mAbs for neutralizing effects (Fig. 2f–k, Supplementary Fig. 5a). All eight mAbs potently neutralized rHIV-NiV_BD, with a half-maximal inhibitory concentration (IC_50) < 0.07 μg ml⁻¹ (Table 1). Seven NiV_MY G cross-binding antibodies neutralized rHIV-NiV_MY with an IC_50 of <0.09 μg ml⁻¹. Four of the five HeV G cross-reactive antibodies demonstrated analogous inhibitory activity against rHIV-HeV, with an IC_50 < 0.04 μg ml⁻¹, while 1D11 failed to neutralize rHIV-HeV (Supplementary Fig. 5a). These four bnAbs were from two groups: 1B6 and 2E7 were in group B, whereas 1E5 and 2A4 were in group C (Fig. 1i). A single amino acid mutation, D582N in HeV G or V507I in NiV G, attenuated m102.4 neutralization[42]. We constructed

rHIV-HeV G_D582N and rHIV-NiV_MY G_V507I strains to test the effects of the two mutants on the neutralizing ability of the bnAbs (Fig. 2i, j). Compared with that of the wild type, the neutralization capacity of the four bnAbs described here did not change markedly (0.5–1.9 fold), whereas the IC_50s of m102.4 against the rHIV-HeV G_D582N variant and the rHIV-NiV_MY G_V507I variant increased by 44.5- and 5.3-fold, respectively (Table 1, Supplementary Fig. 5b). Notably, all the bnAbs powerfully neutralized HeV-g2 (Fig. 2k), a recently discovered genotype of HeV[26]. We confirmed the neutralizing efficacy of the antibody against authentic HNVs through plaque reduction assays (Fig. 2l–n). These pseudovirus-neutralizing antibodies were similarly effective against live viruses (Table 1). Moreover, we compared the neutralizing efficacy of the bnAbs described here against HNVs with that of previously reported G- or F-specific antibodies[15,16,19,43] (Fig. 2o, Supplementary Fig. 5c). Although 1E5 and HENV-26[15] exhibited the best overall HNV

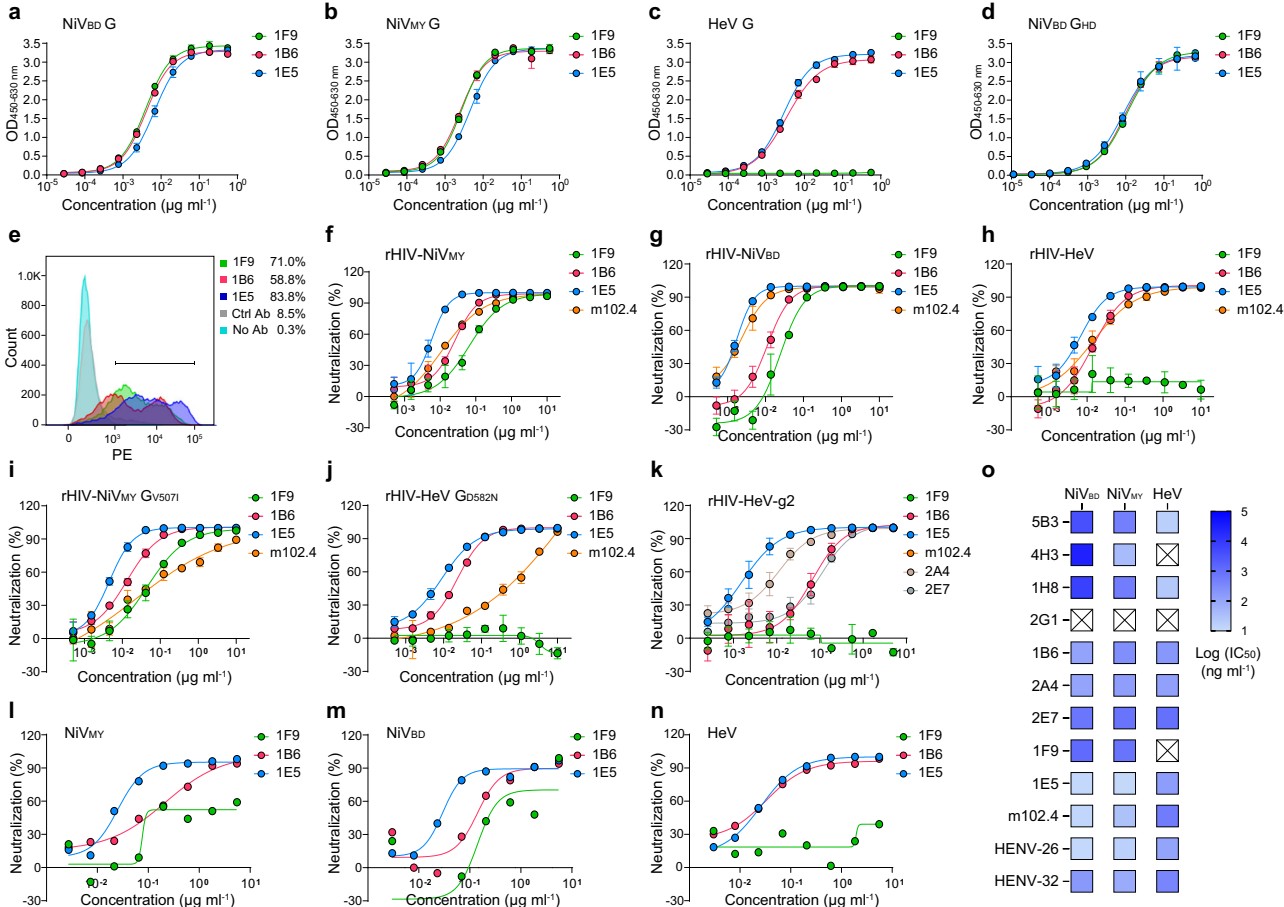

**Fig. 2 | Antibodies exhibit strong cross-reactivity against HNVs.** Reactivity of representative antibodies of each group to the indicated $NiV_{BD}$ G (**a**), $NiV_{MY}$ G (**b**), HeV G (**c**), or $NiV_{BD}$ G head domain (**d**), as determined by ELISA (see also Supplementary Fig. 3a). Data are the mean ± s.d. of three replicates from one representative experiment. **e** Determination of the binding of antibodies to the natural structure of $NiV_{BD}$ G displayed on the 293T cell surface through flow cytometry (see also Supplementary Fig. 3b). Data are presented as the mean of triplicate measurements. Neutralizing curves of antibodies against pseudotyped rHIV-$NiV_{MY}$ (**f**), -$NiV_{BD}$ (**g**), -HeV (**h**), -$NiV_{MY}$ $G_{V507I}$ (**i**), -HeV $G_{D582N}$ (**j**), and -HeV-g2 (**k**), (see also Supplementary Fig. 5a). Data are presented as the mean ± s.d. of three replicates from one representative experiment. Neutralizing curves of antibodies against authentic $NiV_{MY}$ (**l**), $NiV_{BD}$ (**m**), or HeV (**n**). Representative data from one test are shown. **o** The neutralizing efficacy of representative G or F antibodies against HNVs (see also Supplementary Fig. 5b). Source data are provided as a Source Data file.

cross-neutralization ability, their inhibitory activity against HeV was lower than that of two F-targeting antibodies.

## Analysis of neutralizing mechanisms

Because rHIV-HNVs infect only one round, the neutralization of mAbs should be due to inhibiting the cellular entry of virions. HNV cell entry is a complex multistep process involving the binding of G to host receptors, the activation and structural rearrangement of F, membrane fusion, and the release of the viral genome[13,16]. The presence of mAbs from different groups inhibited syncytium formation (Fig. 3a), suggesting that these G-specific antibodies prevent virus invasion before membrane fusion. To analyze the neutralization mechanism of the mAbs, we produced recombinant human EB2/EB3-Fc chimaeras (Supplementary Fig. 2b). We then examined the ability of the mAbs to block EB2/EB3-Fc binding to HNV G proteins using competitive Luminex assays. NiV G-specific antibodies in group A failed to block receptor binding (Fig. 3b, Supplementary Fig. 6a). Antibodies from group B only partially blocked receptor binding to NiV G at high concentrations ($IC_{50} > 2.0\ \mu g\ ml^{-1}$), while most mAbs in group C exhibited potent blocking ability ($IC_{50} < 0.2\ \mu g\ ml^{-1}$). For HeV G, only bnAbs 1E5 and 2A4 effectively blocked receptor binding. The average blocking capacities of the mAbs against $NiV_{BD}$ G binding to EB2 and EB3 were 2.2 and 1.8 times greater than that of $NiV_{MY}$ G, respectively (Supplementary Fig. 6b), which may explain the similar binding and neutralizing titres

of serum to the two NiV G proteins (Fig. 1b, c), but the different inhibitory effects on the receptor (Fig. 1d). We verified the ability of the mAbs to block EB2 from binding to $NiV_{BD}$ G using BLI assays, and similar results were obtained (Supplementary Fig. 6c).

We sought to determine whether the mAbs retained their blocking activity when the receptor or G was anchored to the membrane surface. EB2, considered more active in HNV adhesion[44], was used for flow cytometry analyses. Preincubation with representative antibodies from groups A (1F9) and B (1B6) or an irrelevant antibody (2G1[40]) did not block the binding of soluble EB2 to membrane-displayed NiV G or HeV G, whereas bnAb 1E5 in group C partially blocked EB2 binding (Fig. 3c, d, Supplementary Fig. 6d). The blocking activity of 1E5 and m102.4 was weaker than that of EB2-Fc itself, possibly owing to their different spatial accessibilities to the lower two heads of the G-tetramer on the membrane surface, because EB2-Fc seems to be smaller in size (Supplementary Fig. 2b). When full-length EB2 was present on the membrane, 1E5 completely blocked the binding of G at low concentrations (Fig. 3e, Supplementary Fig. 6d).

The receptor and G-tetramer are confined to the membrane surface under natural circumstances. The binding of non-RBD-targeted antibodies in groups A and B may alter the structure of the G-tetramer or generate steric hindrance, preventing virions from adhering to the surface of the host cell. We used h5B3.1, a bnAb that targets the prefusion conformation of HNV F[16], to verify this hypothesis. Compared

**Table 1 | Summary of the binding and neutralizing activity of nAbs**

| mAb | Epitope Group | ELISA binding EC$_{50}$ values (ng ml$^{-1}$) | | | | PPCs (%) | Kinetic constant (nM) determined by BLI | | | Pseudovirus neutralization IC$_{50}$ values (ng ml$^{-1}$) | | | | | Live virus neutralization IC$_{50}$ values (µg ml$^{-1}$) | | |
|---|---|---|---|---|---|---|---|---|---|---|---|---|---|---|---|---|---|
| | | NiV$_{BD}$ G | NiV$_{MY}$ G | HeV G | NiV$_{BD}$ G$_{HD}$ | NiV$_{BD}$ G | NiV$_{BD}$ G | NiV$_{MY}$ G | HeV G | NiV$_{BD}$ | NiV$_{MY}$ | HeV | NiV$_{MY}$- V507I | HeV-D582N | NiV$_{BD}$ | NiV$_{MY}$ | HeV |
| 1A9 | A | 6.5 | NM | NM | 7.0 | 80.2 | 0.02 | NM | NM | 7.8 | NM | NM | NM | NM | 0.09 | NM | NM |
| 1F9 | A | 3.9 | 2.7 | NM | 10.9 | 71.0 | 0.01 | <0.01 | NM | 32.4 | 68.9 | NM | 52.1 | NM | 0.26 | 0.1 | NM |
| 1B6 | B | 4.0 | 2.5 | 3.4 | 9.4 | 58.8 | <0.02 | <0.02 | 0.56 | 12.3 | 22.8 | 16.5 | 12.0 | 20.2 | 0.14 | 0.13 | 0.17 |
| 2E7 | B | 6.6 | 6.4 | 9.4 | 18.8 | 83.5 | <0.10 | 0.24 | <0.04 | 69.7 | 87.2 | 34.0 | 59.3 | 42.3 | 1.52 | 0.62 | 0.61 |
| 2A4 | C | 6.1 | 4.4 | 12.1 | 5.0 | 77.1 | <0.02 | <0.01 | 0.08 | 4.7 | 6.3 | 5.5 | 12.0 | 6.4 | 0.45 | 0.47 | 0.92 |
| 1D11 | C | 4.4 | 3.0 | 33.2 | 9.4 | 75.5 | 0.43 | <0.01 | 3.66 | 2.3 | 5.7 | NM | 5.6 | NM | < | 0.02 | 29.0 |
| 1E5 | C | 6.6 | 4.5 | 2.8 | 8.6 | 83.8 | 0.05 | <0.01 | <0.01 | 1.7 | 5.1 | 5.7 | 4.3 | 7.4 | 0.03 | 0.03 | 0.19 |
| 2B8 | C | 2.1 | 2.8 | NM | 5.3 | 80.3 | <0.01 | 0.05 | NM | 3.8 | 22.6 | NM | 16.6 | NM | 0.02 | 0.07 | NM |
| m1O2.4 | C | 8.7 | 6.7 | 5.8 | 6.1 | 80.2 | ND | ND | ND | 2.1 | 15.0 | 14.3 | 79.3 | 636.1 | ND | ND | ND |

ELISA binding and neutralization experiments were performed at least twice, and similar results were obtained. The half-maximal effective concentration (EC$_{50}$) or half-maximal inhibitory concentration (IC$_{50}$) was calculated using GraphPad Prism 8.0. PPCs, the proportion of phycoerythrin-positive cells according to flow cytometry, as determined in Supplementary Fig. 3b. NM, not measurable. ND, not done. The "<" in affinity measurements indicates that the koff value was calculated as 1×10$^{-6}$ due to the low dissociation rate. The "<" in the neutralization assays indicates that an IC$_{50}$ above the lowest concentration of 0.003 µg/mL was not achieved for NiV$_{BD}$.

with the control mAb, 1E5 markedly blocked the adhesion of the rHIV-NiV$_{BD/MY}$ particles (Fig. 3f, g, Supplementary Fig. 6d). 1F9 and 1B6 did not prevent virions from binding to the receptor, suggesting that they neutralize by inhibiting G conformational changes or G-F interactions. Collectively, these mAbs possess different neutralizing mechanisms and have the potential to act synergistically or constitute a cocktail therapy.

### In vivo protection against NiV in hamsters

We selected the most potent neutralizers in each epitope group to evaluate their pre- and post-exposure protective efficacy against NiV infection in hamsters. Hamsters were infected with a lethal dose of NiV$_{MY}$, followed by intraperitoneal (i.p.) administration of phosphate buffer solution (PBS) or a single dose of each nAb one day before or after infection (dpi). Significant protection was observed in the low-dose (4.5 mg/kg) mAb intervention group, except for the group administered 1B6 at 1 dpi (Fig. 3h). In particular, the prophylactic administration of 1F9 or therapeutic administration of 1E5 achieved full protection. We further evaluated the protective capacity of the mAbs at a higher dose (10 mg/kg). All hamsters pretreated with 1E5 survived the challenge ($P = 0.0005$) without obvious weight loss or disease symptoms (Fig. 3i). 1B6 also conferred significant protection when administered one day before ($P = 0.0038$) or after ($P = 0.0005$) infection. Notably, both doses of 1B6 failed to achieve high survival rates at 1 dpi but significantly extended the survival time of the dead hamsters. These results suggest that mAbs from different epitope groups are effective protective agents against HNV infection.

### Crystal structure of NiV$_{BD}$ G$_{HD}$ in complex with the bnAb 1E5

To understand the structural basis of the most potent bnAb from the panel, we determined the crystal structure of the 1E5 Fab in complex with the NiV$_{BD}$ G$_{HD}$ at 3.24 Å resolution (Fig. 4a, b, Supplementary Fig. 7a). 1E5 binds to the central cavity of the G head at an angle nearly perpendicular to the plane formed by the six blades of the β-propeller. The high shape complementarity between the CDRs of 1E5 and the central cavity region of the G$_{HD}$ facilitates extensive residue contacts between them (Fig. 4c). There are thirty-one residues from CDRH1-3 or CDRL1-3 and two additional residues from FR-L3 involved in the interaction with the 28 residues around the central cavity of G to form 18 hydrogen bonds (H-bonds) at the contact interface (Fig. 4d), providing the basis for the high affinity of 1E5. Remarkably, the long, protruding CDRH3 of 1E5 is deeply inserted into the central cavity and contributes half of the H-bonds at the 1E5-G interface (Fig. 4e). We compared the complex structures of 1E5 and m102.3 (an m102.4 derivative) (PDB: 6CMI) with that of EB2 (PDB: 2VSM) (Fig. 4f, g). The burial surface areas of the EB2, 1E5, and m102.3 epitopes on the G head are 1406.5 Å$^2$, 1736.2 Å$^2$, and 1057.1 Å$^2$, respectively (Supplementary Fig. 7b–d). Although the epitopes of both RBD-targeting antibodies highly overlap with those of EB2 (Fig. 4h) and their CDRH3s act in a way similar to the G-H loop of EB2[45] (Supplementary Fig. S7e), there are still some differences in the details of the binding modes. Compared to 102.3, 1E5 has almost the same binding orientation as EB2 (Fig. 4f, g), a larger EB2 epitope coverage area (Fig. 4h), and a deeper insertion depth of CDRH3 to the central cavity (Supplementary Fig. S7e).

Analysis of 1E5 epitope conservation between NiV and HeV G revealed that residues buried by 1E5 are strictly conserved (Supplementary Fig. S7f). We chose the residues likely to interact with 1E5 to construct single-site mutants of G and analyzed the changes in the binding activity of antibodies or EB2 to these mutants (Fig. 4i, j). Residues W504, E579, T507, G506, Y458, and L305 around the central cavity were identified as the key sites for 1E5 binding (Fig. 4i). In addition to L305, these sites also play an important role in the binding of G to EB2 (Fig. 4j). Notably, mutations in CDRH3 did not affect the binding of 1E5 to NiV or HeV G (0.4- to 2.2-fold change in EC$_{50}$) (Fig. 4k),

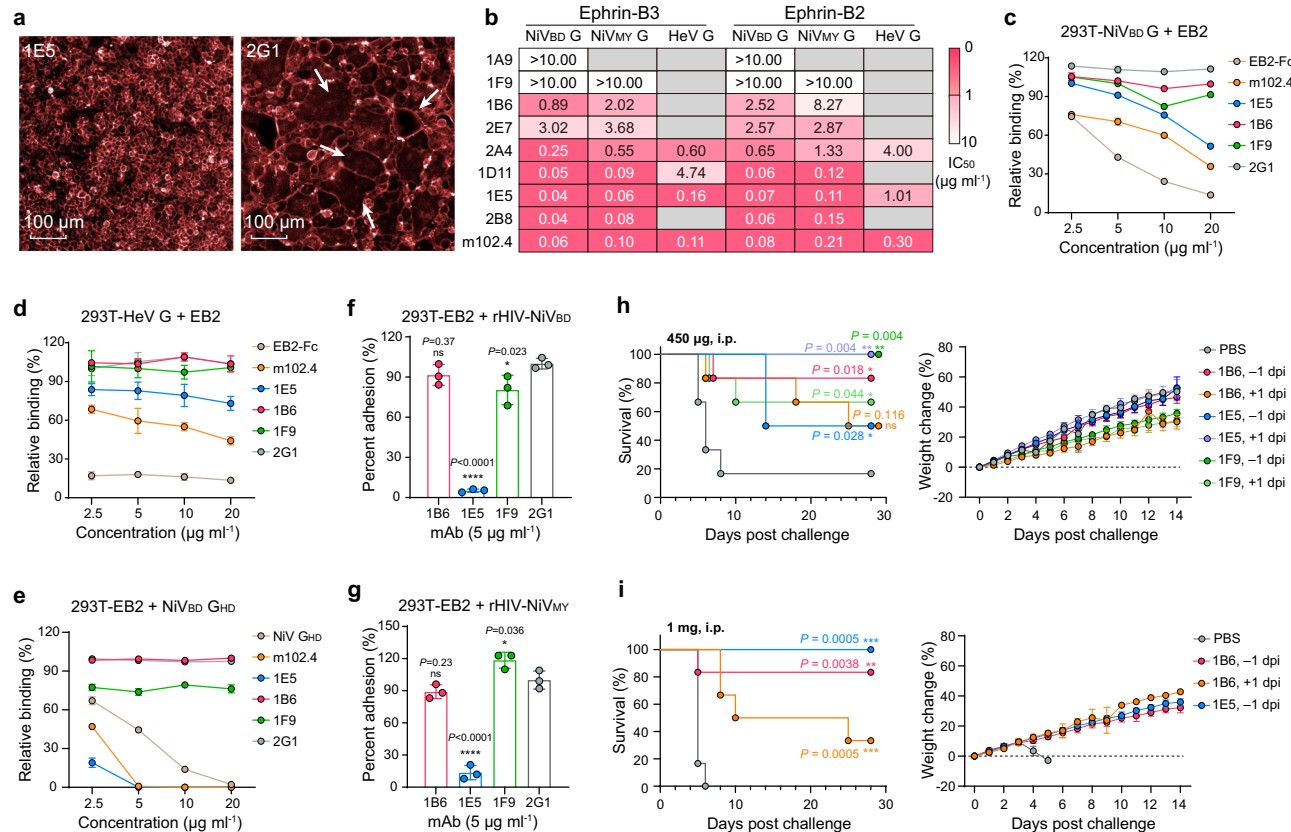

**Fig. 3 | Protective antibodies have multiple neutralization mechanisms. a** Inhibition of fusion in 293T cells cotransfected with plasmids encoding full-length of G and T5F by antibodies. Arrows point to typical syncytia. **b** The ability of antibodies to block the binding of recombinant receptors to soluble G ectodomains was tested by Luminex assays. The magenta to white gradient grid indicates an IC₅₀ value ranging from 0 to 10 µg ml⁻¹, and gray grids indicate that no blocking activity was detected (see also Supplementary Fig. 5b). The ability of antibodies to block EB2-Fc (**c**, **d**) or NiV_BD-G_HD (**e**) binding when full-length NiV_BD (**c**)/HeV G (**d**) or ephrin B2 (**e**) was anchored on the 293T membrane. The relative binding percentage was calculated as the ratio of positive cells in the presence or absence of antibodies. EB2-Fc (**c**, **d**) or NiV_BD-G_HD (**e**) was used as a positive control. Data are presented as the mean ± s.d. of three replicates. The ability of antibodies to block the adhesion of rHIV-NiV_BD (**f**) or rHIV-NiV_MY (**g**) virions (mean ± s.d. of triplicate

measurements). In Dunnett's multiple comparisons test, the mean difference (MD), 95% Confidence interval (CI), and P value were [94.87, 78.35–111.4, P < 0.0001] (**f**) and [86.47, 69.35–103.6, P < 0.0001] (**g**) for 1E5, [19.6, 3.08–36.12, P= 0.023] (**f**) and [−18.5, −35.62 to −1.38, P= 0.036] (**g**) for 1F9, and [8.5, −8.02 to 25.02, P= 0.371] (**f**) and [10.9, −6.22 to 28.02, P= 0.232] (**g**) for 1B6, respectively. *P < 0.05, **P < 0.01, ***P < 0.001, ****P < 0.0001, ns, not significant. The EBOV-specific antibody 2G1 was used as an isotype control (**c**–**g**). **h**, **i** Hamster protection studies. Kaplan–Meier survival curve (left) and weight change (right, data are presented as mean ± s.d.) of hamsters treated with 450 µg (**h**) or 1 mg (**i**) of antibody (n = 6). The P value and the asterisk indicate statistical significance determined by a Mantel-Cox log-rank test. The color of the survival curve for each antibody matches that of the weight change curve on the right of the chart. Source data are provided as a Source Data file.

which may be related to the widespread involvement of the other CDRs (Fig. 4c, d). We also evaluated the potential effect of oligosaccharides on the 1E5-G interaction. We docked five N-linked glycans in the head domain[35] into our model and found that only N306 and N529 were located near the epitopes of 1E5 (Supplementary Fig. S7g). The average minimum distance between the N-acetylglucosamine of N306 and the surface of the 1E5 Fab is approximately 7 Å, whereas that between N529 and the 1E5 Fab is approximately 15.7 Å. In this case, glycosylation of N306 and N529 should not interfere with the binding of 1E5. In summary, as a receptor-like antibody, the 1E5 epitope is strictly conserved on HNV G proteins and highly overlaps with receptor-binding sites, laying the foundation for its broad neutralization of HNVs.

**Dynamic structures of the NiV G-tetramer induced by receptor-like 1E5 Fab**
The binding of receptors or RBD-targeting antibodies causes only minor structural changes in the G head[15,33,34,42], with root-mean-square deviations (RMSDs) ranging from 0.41–0.65 Å for 413–427 Cα atoms (Supplementary Fig. S7h). We sought to use 1E5 Fab, which has a binding mode highly similar to that of the receptor (Fig. 4), to analyze

the potential conformational dynamics of the G-tetramer after receptor engagement necessary for membrane fusion. First, we examined the structural correctness of the soluble NiV_MY G ectodomain through negative-stain electron microscopy, and confirmed that the homo-tetramer has a comparable quaternary structure to that previously solved with nAH1.3-Fab[35] (Supplementary Fig. 8a). We then performed structural studies on the G-tetramer-1E5 Fab complexes by cryo-EM. We observed two dominant antibody-antigen complex types during data processing, each containing two heads with intact high-density (Supplementary Fig. 8b, c). These complexes differed significantly in the flexibility of the overall architecture and thus were designated the loose (Fig. 5a) or compact (Fig. 5b) types. Some explicit images at different orientations in the 2D classification allowed us to determine that low-density parts are the remaining heads of the G-tetramer despite their differences in sizes and details from the well-determined heads in the final models (Fig. 5a, b, Supplementary Fig. 8b, c). However, the flowability in the spatial position of the mobile heads made them weakly resolved, and we finally determined the incomplete 1E5/G-tetramer structures of the upper heads of the loose type or the lower heads of the compact type at resolutions of 2.94 Å (Fig. 5c) and 3.18 Å (Fig. 5d), respectively.

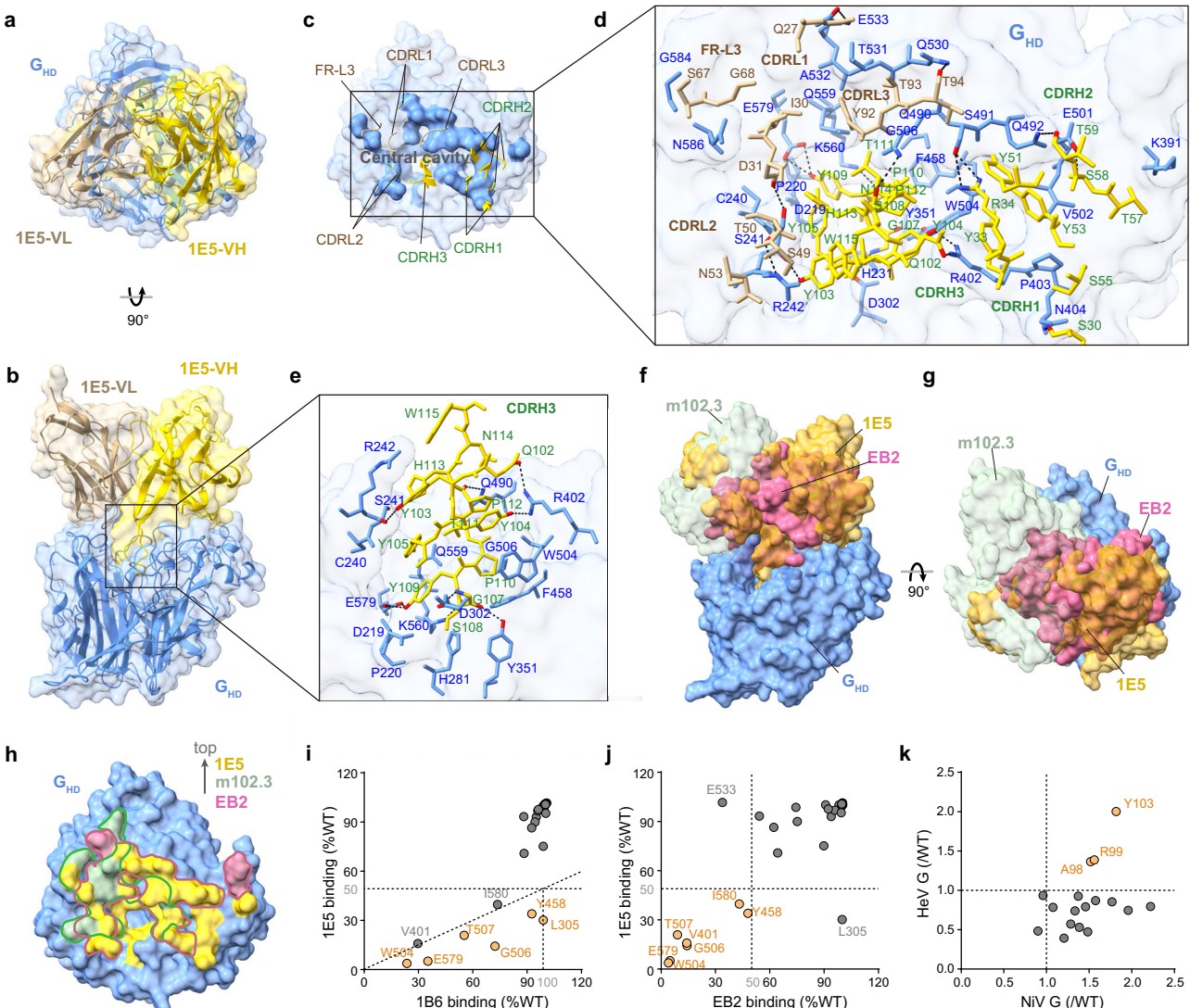

**Fig. 4 | 1E5 binds to the G head domain in a way that mimics the receptor.** Top (**a**) and side (**b**) views of the crystal structure of the NiV$_{BD}$ G$_{HD}$/1E5 Fab complex. Molecules are shown as ribbon diagrams beneath a transparent surface. The G$_{HD}$ is colored cornflower blue, and the V$_H$ and V$_L$ of 1E5 are shown in gold and tan, respectively. **c** Top view of the interface between NiV G and 1E5. The NiV G$_{HD}$ is represented as a molecular surface, and the epitopes of 1E5 are highlighted. The paratopes of 1E5 are shown as ribbons. **d** Residues and H-bonds at the interface of the NiV G/1E5 complex. Contact residues are shown as stick representations, and H-bonds are shown as dotted black lines. CDRH1-3 is colored gold, CDRL2-3 and FR-L3 are colored tan, the oxygen atoms are colored red, and the nitrogen atoms are colored bright blue. **e** Interaction of 1E5 CDRH3 with the NiV G central cavity. NiV G is shown as a transparent surface representation, and the other elements are shown as described above. **f**–**h** Superimposition of the G$_{HD}$/1E5 complex with the G$_{HD}$/EB2 (PDB ID: 2VSM) and G$_{HD}$/m102.3 (PDB ID: 6CMI) structures. Molecules are shown as surface representations. The G$_{HD}$, 1E5, EB2, and m102.3 are colored cornflower blue, gold, pale violet red, and dark sea green, respectively. The epitopes of m102.3, EB2, and 1E5 are superimposed in their parent colors on the G$_{HD}$ surface. **i, j** Relative binding activity of antibodies or EB2 to the mutants and wild-type HeV G. Residues possibly critical for 1E5 binding were mutated to alanine, and the ratio of the area under the curve tested by ELISA was calculated. **k** Relative binding activity of the 1E5 CDRH3 alanine mutants to HNV G. The ratio of the EC$_{50}$ of the 1E5 mutants to that of the wild-type was calculated. Source data are provided as a Source Data file.

In the loose type, the binding of 1E5 to the membrane-distal heads appears to destroy the stability of the tetramer, which results in apparent shifts of the membrane-proximal heads relative to the stalk bundle (Fig. 5a, c). In contrast, in another type, the observable 1E5 binds to the lower two heads at an angle approximately parallel to the stalk to form a pseudo-symmetrical structure with a relatively small overall change in the G-tetramer (Fig. 5b, d, Supplementary Fig. 8c).

Similarly, we found no noticeable change in the head domains between the two types of structures or when compared with the structure defined by crystallography, with a maximum RMSD value of 0.631 Å (Fig. 5e). When superimposed with the corresponding two protomers of previously reported G-tetramer, the compact type

manifested a high degree of coincidence (Fig. 5f), while the loose type exhibited a moderate difference (Fig. 5g). The head A of the loose structure rotates counterclockwise by approximately 23° around the center of the two upper heads, resulting in changes in residue interaction at the interface of heads A and B (Fig. 5g). Only two residues, Y205 and R258, of the original seven residues of head A remained on the interface and generated spatial displacements of 17.0 Å and 4.2 Å, respectively. Moreover, the two H-bonds formed by the Nε and Nη of R258[B] and the main-chain oxygen of L202[A] (D-A distances of 3.23 Å and 2.18 Å, respectively) replaced the original two H-bonds, S204[A]-G259[B] and K591[A]-E261[B]. These changes may lead to the opening of the interlaced β-sandwich at the neck region to release the membrane-proximal heads.

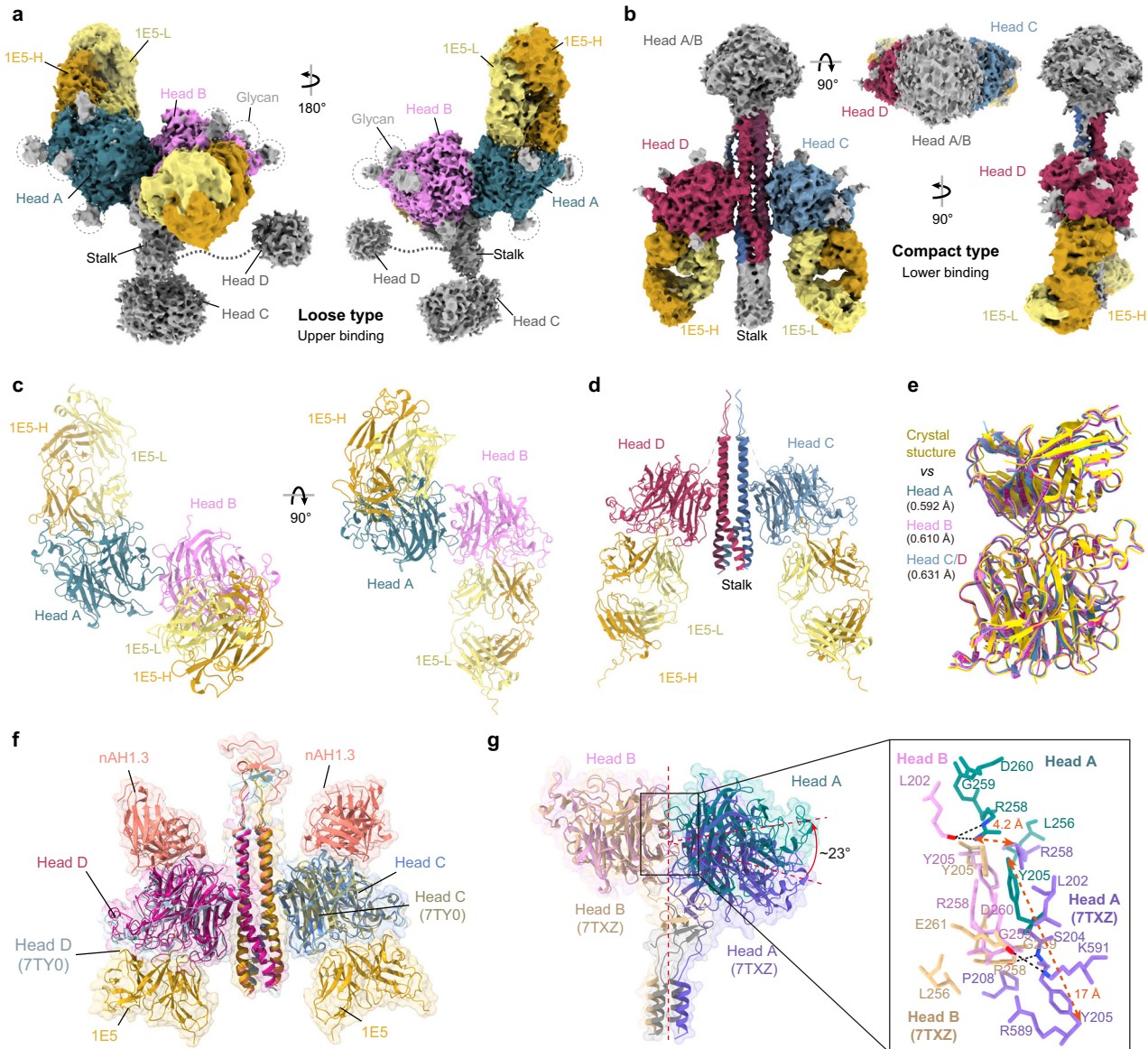

**Fig. 5 | 1E5 induces two distinct dynamic structures of the G-tetramer.**
**a** Molecular surface showing two flip-horizontal views of the upper binding loose type of the NiV/1E5 complex. The glycans are marked with gray dotted circles, the two unsolved lower heads are shown in gray, and the other chains are shown in different colors as indicated. **b** The approximate symmetric structure of the lower binding compact type of the NiV/1E5 complex, represented as a molecular surface. Structural diagram of 1E5 Fabs complexed with the upper two heads of the loose type (**c**) or the lower two heads of the compact type (**d**), shown in colored ribbon representation. **e** The superimposition of G_HD-1E5 structures determined by cryo-

EM and crystallography. The superimposition of the compact (**f**) or loose type (**g**) with the G-tetramer previously reported (PDB ID: 7TXZ/7TY0). Local residue contacts are shown as stick representations in their parent chain colors, and H-bonds are shown as dotted black lines. Heads A, B, C, and D of 7TXZ/7TY0 are colored tan, light medium purple, dark khaki, and light blue, respectively. Heads A, B, C, and D of the loose/compact type are colored teal, violet, cornflower blue, and medium violet red, respectively. 1E5 is colored goldenrod, and nAH1.3 is in salmon. The red dotted line with a bidirectional arrow indicates the offset distance.

## Identification of critical sites for G triggering and F activation

A previous study indicated that the mobile stalk of G triggered the conformational change of F[14]. However, in both that and the present study, cotransfection of full-length or the truncated stalk of G with F inefficiently induced fusion and did not form canonical syncytia (Supplementary Fig. 9a). Therefore, we speculate that there may be other trigger sites that are more crucial. To identify the trigger sites of G required for F activation, we used nickel-coated plates to capture prefusion soluble F (sF) to simulate its natural orientation on the membrane and then searched for potential action sites of G using 148 overlapping peptides spanning the entire NiV_BD G. Six peptides were obtained by preliminary screening (Supplementary Fig. 9b), among which P-59 (233–247 aa) was further identified to block the binding of

h5B3.1 to sF significantly (Fig. 6a). We employed microscale thermophoresis (MST) and BLI to further confirm the mutual effect of P-59 with sF, with affinity $K_D$ values of $5.0 \pm 1.3$ µM and $0.19 \pm 0.14$ µM, respectively (Fig. 6b, Supplementary Fig. 9c). When mapped onto the G head, P-59 forms the extended outer loop β1S2-β1S3 (Supplementary Fig. 9d), which is located on the upper edge of the central cavity[34]. In the G-tetramer, four β1S2-β1S3 loops are located on the side of the head near the tetrameric central axis (Supplementary Fig. 9e), which is difficult for nearby F to contact. Not surprisingly, neither the extracellular nor the head domain of G affected the binding of h5B3.1 (Supplementary Fig. 9f), suggesting that both conformational changes and the presence of stalks may be necessary for F triggering.

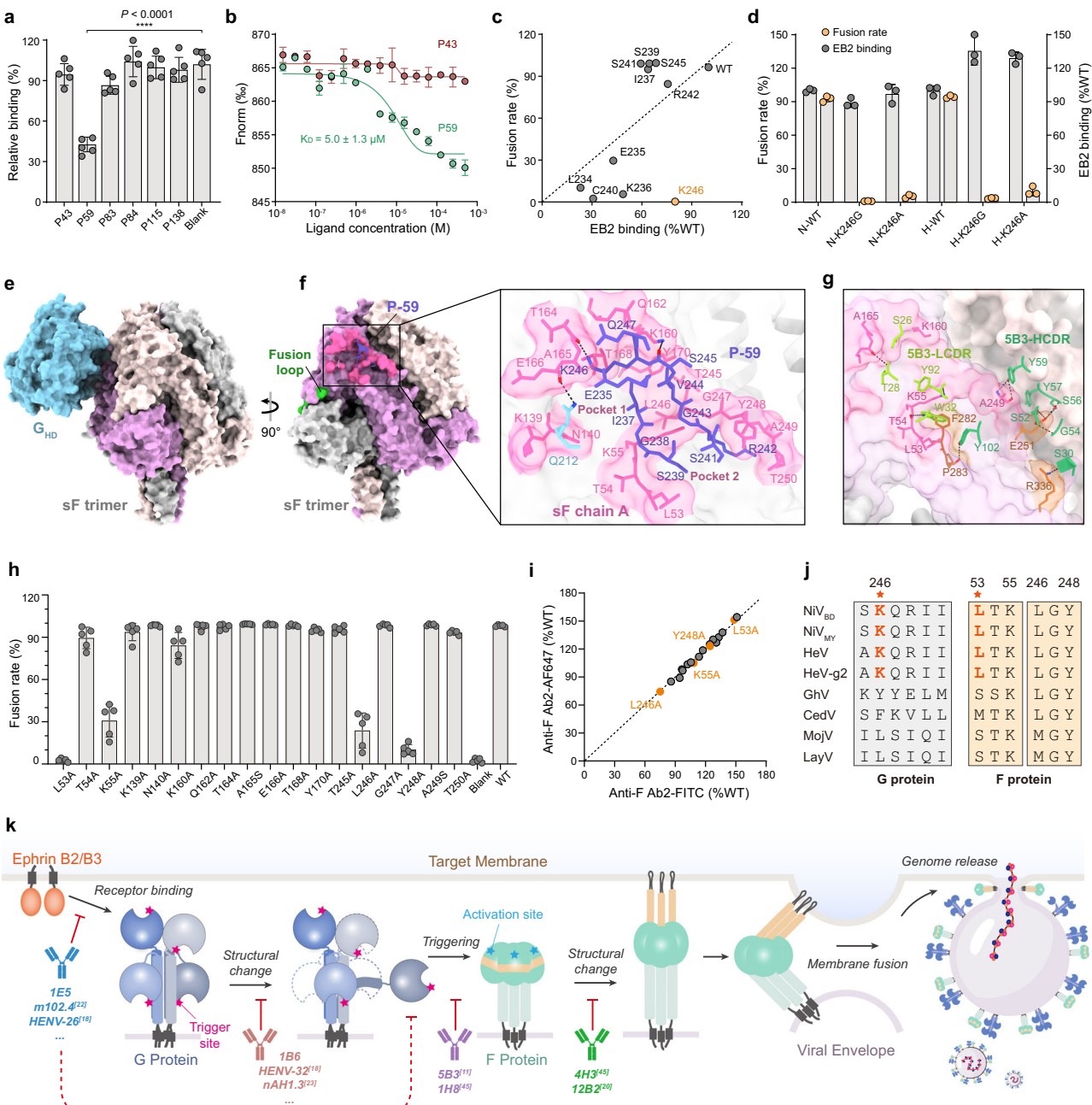

**Fig. 6 | The extended outer loop β1S2-β1S3 of G inserts into the two pockets on the apical region of F to trigger fusion. a** Validation of the preliminary screened reactive peptides. Data are presented as mean ± s.d. of five replicates from one test. In Dunnett's multiple comparisons test, the MD, CI, and P values between P59 and control were 58.17, 43.25 to 73.10, and $P < 0.0001$, respectively. **b** Interaction affinity between P-59 and sF determined by microscale thermophoresis. The experiments were performed twice, and similar results were obtained. Data are presented as mean ±s.d. of three replicates from one representative experiment. **c** Fusion rate and relative EB2 binding of P-59 related single-site mutants (see also Supplementary Fig. 9h, i). Data are presented as the means of three replicates from one representative experiment. **d** Fusion rate and EB2 relative binding of wild type or K246G/A mutants of NiV or HeV G. Data are presented as mean ±s.d. of three replicates from one representative experiment. **e** Side view of the top 1 pose of docked $G_{HD}$-sF structures. Molecules are shown as surface representations. $G_{HD}$ is colored sky blue,

and the chains A/B/C of sF are colored plum, light gray, and misty rose, respectively. **f** The zoomed view of the interface between $G_{HD}$ and sF. P-59 is shown as a ribbon diagram colored slate blue. The epitope of P-59 is highlighted and colored hot pink. Contact residues are shown as stick representations in parent colors, and H-bonds are shown as dotted black lines. **g** Contact residues at the interface of 5B3 and sF. The HCDR and LCDR of 5B3 are shown in dark and light green, respectively. Epitopes specific to 5B3 are shown in brown, and the residues shared by 5B3 and P-59 are shown in hot pink. **h** Fusion rate of the T5F mutant cotransfected with full-length of G. Data are presented as mean ±s.d. of five replicates from one representative experiment. **i** Relative binding of the mutants and wild-type of NiV T5F to two noncompetitive F-specific antibodies (see also Supplementary Fig. 9l). Data are the means of three replicates from one representative experiment. **j** Conservation of critical residues for G-F interactions among henipaviruses. **k** A new model for virus invasion and antibody neutralization. Source data are provided as a Source Data file.

Next, we constructed single point mutants of G and found that the L234–K236, C240, and K246 mutations strongly inhibited syncytium formation when the cells were cotransfected with F (Fig. 6c, Supplementary Fig. 9g). Considering that mutations may interfere with

receptor binding by causing G expression or conformational impairment, thereby affecting membrane fusion, we evaluated the structural correctness of these variants. The K246G mutation had a greater proportion of EB2-binding G proteins than the other mutants (Fig. 6c,

Supplementary Fig. 9h) and did not hinder the binding of antibodies targeting different epitopes (Supplementary Fig. 9i), but eliminated syncytium formation (Fig. 6c, Supplementary Fig. 9g), suggesting that it was a key trigger site for the G/F interaction. Mutation of the conserved K246 residue to glycine or alanine in HNV G similarly reduced cell–cell fusion while retaining the binding ability of EB2 (Fig. 6d). Since the β1S2-β1S3 loop is spatially adjacent to receptor binding sites (Supplementary Fig. 9d, e), mutations may lead to changes in receptor recognition epitopes. Therefore, although the mutation resulted in fewer EB2-positive cells, it almost entirely blocked syncytium formation (Fig. 6c), indicating that sites such as K236 and C240 may also be involved in the triggering process.

To identify activation sites on F, we performed rigid body docking of $G_{HD}$ and sF using ZDOCK[46] in Discovery Studio 4.5 (Supplementary Fig. 9j). P-59 was restricted to contact with the sF, and the pose with the highest score among all the clusters was subjected to ligand-receptor interaction analysis. In the top 1 pose, P-59 and an additional residue, Q212, recognize 18 residues located at the periphery of two pockets within apical domain 3 of sF (Fig. 6e, f). In particular, six residues, L53, T54, K55, K160, A165, and A249, of two pockets participate in the interaction with 5B3, a neutralizing antibody that inhibits F-mediated fusion by locking F in the prefusion state (Fig. 6g, Supplementary Fig. 9k)[16]. Next, we constructed single-point mutants of T5F and evaluated their effects on membrane fusion. Mutations in residues L53, K55, T245, and Y248 significantly prevented syncytial formation (Fig. 6h). We confirmed that these mutations do not affect F expression or conformation (Fig. 6i, Supplementary Fig. 9l) using two non-competitive anti-F antibodies, suggesting that they may be important fusion activation sites of F. We compared the conservation of the identified critical sites of G-F interactions within the genus Henipavirus and found that both K246 of G and L53 of F are conserved only in HNVs (Fig. 6j), which may be one of the reasons why HNVs differ from other species in invasion mechanism and pathogenicity.

Based on the above analysis, we updated the model of HNV invasion and antibody neutralization (Fig. 6k). Receptor binding (i) leads to dynamic structural changes in the G-tetramer (ii) to expose the trigger sites on the lower head. The pockets within the apical region of F receive the trigger signal (iii) and activate the conformational change (iv) required for membrane fusion. All of these processes can be blocked by reported antibodies to neutralize the virus.

## Discussion

The high virulence, zoonotic potential, wide host range, and broad geographical distribution of HNVs make them a great threat to humans. However, no approved prophylactic or therapeutic measures for human use are available[47]. Although multiple prevention or treatment options are promising[2,15,19,22,48–51], additional candidates need to be developed to cope with the ever-changing viruses.

Consistent with the findings of a previous study[38], we found that homologous G alone failed to induce a comparable humoral response to HNVs (Fig. 1b, c). Both NiV- and HeV-specific antibodies increased in the serum after HeV G booster vaccination; however, neither the NiV nor HeV G RBD-targeted antibodies were further promoted (Fig. 1d). This was confirmed by the limited cross-receptor-blocking ability of the isolated bnAbs (Fig. 3b). Epitope bias may exist during booster vaccination with heterologous G, and antibodies targeting the more dominant epitope of HNV G, other than the RBD, may be further stimulated. In addition, several antibodies isolated from an HeV-sG-immunized individual enhanced receptor binding or could not access the epitope of tetrameric G[6]. Despite these results being attributable to immune response differences in individuals or among species, optimized immune strategies or rational antigen designs (e.g., retaining/stabilizing the structure of the G-tetramer or exposing the conserved epitopes of the RBD) may be necessary to arouse a more desirable broad immune response or bnAbs against HNVs.

The application limitations of vaccines in particular situations and populations and the unsatisfactory efficacy and side effects of small antiviral drugs highlight the importance of developing therapeutic antibodies[12]. Limited antibodies under development are not sufficient to fight against these dangerous viruses[7–15]. Here, our detailed characterization of a panel of antibodies against HNVs can provide valuable insights for the development of immunotherapies for HNV infections. i) The identified antibodies can potently neutralize HNVs and have significant in vivo protective efficacy (Figs. 2a–n and 3h, i), making them promising drug candidates for HNVs. ii) The nAbs recognize multiple epitopes (Fig. 1i) and possess different neutralizing mechanisms (Fig. 3b–g), giving them the potential to work together in a cocktail to achieve better protection against HNVs through synergistic or complementary effects. iii) Three bnAbs, 1B6, 1E5, and 2A4, which have the most potent broad neutralizing capacity (Table 1), possessed longer CDRH3s (19–21 aa) than other mAbs (9–14 aa) and shared the same J gene, IGHJ5 (Fig. 1h). The combination of long CDRH3 and IGHJ5 may represent a class of effective macaque bnAbs against HNV infections. We speculated that long CDRH3 may facilitate antibodies to improve structural flexibility and epitope accessibility, and the choice of IGHJ5 may help stabilize the CDRH3 loop and contribute to the paratope. iv) The extensive participation of all CDRs of 1E5 (Fig. 4c, d) and its high consistency with the critical residues for receptor binding (Fig. 4f–j) may increase its tolerance to viral escape mutations. v) We observed differences in the neutralizing ability of G and F antibodies against NiV and HeV (Fig. 2o), which may be related to the abundance of the two glycoproteins on the virion surface, suggesting that the combination of G and F antibodies may be a better choice for treating HNV infections.

Antibodies targeting the RBD are more likely to cross-react and strongly neutralize[6,15,20]. Nevertheless, previous structural analyses were limited to the acquisition of epitopes from antibodies/G-monomers and lacked a holistic interpretation of the interaction between these antibodies and the G-tetramer. We analyzed the structure of the intact G-tetramer ectodomain using the receptor-like bnAb 1E5 and found that the binding of 1E5 induced the G-tetramer to produce two dynamic structures (Fig. 5a, b)[35]. The ultimate structural type should be associated with the head to which the antibody preferentially binds. In the free state, the smaller Fabs should have the same probability of contacting the four heads. However, given that the lower two heads are close to the membrane under natural conditions, it may be difficult for the entire 1E5 to access the downward epitope due to steric hindrance. This was supported by the relatively weak blocking activity of 1E5 against soluble EB2 binding to anchored G (Fig. 3c, d). The dominance of upper binding may be beneficial because blocking receptor seems to be the most efficient[6,15,20], and receptor-like antibodies will not be consumed on the lower heads, which may be one of the reasons for the lower $IC_{50}$ and better efficacy of this type of antibody.

HNV entry involves host receptor-induced spatiotemporally sequential conformational cascades of G and F[52]. A recent study provided a new understanding of how the G stalk affects G/F interactions and membrane fusion[53]. However, the mechanism by which the G-tetramer activates F-trimer remains unclear. Since receptor-like antibodies or receptor binding cause only minor conformational changes in the bound head (Supplementary Fig. S7h)[15,42], the trigger signal is likely generated by the stalk or other heads. Here, we determined that aa 233–247, which form the extended outer loop, β1S2-β1S3, of G, interact strongly with sF and identified that the conserved K246 in this region is critical for F triggering (Fig. 6a–d). The β1S2-β1S3 loop in the G-tetramer is naturally inaccessible to F. Nevertheless, we found that the upper binding of 1E5 caused significant changes in the G-tetramer. Given the high similarity of binding epitopes and modes between 1E5 and the receptor (Fig. 4), assuming that the receptor induces structural changes in the G-tetramer similar to those in the upper bound 1E5 is reasonable. Changes in the position or structure of

the three RBD-downward heads after receptor binding, especially the movement of the lower heads (Fig. 6a, k), may provide opportunities for trigger site exposure and contact range increase. Although only the RBD up-oriented head of the G-tetramer may be easily obtained for pairwise ephrins on the host cell membrane and the G-H loop is not inserted into the central cavity as deep as the 1E5 CDRH3, the constraint and mobility of the stalk and the mechanical force between the virion and the host cell may provide additional driving forces for structural changes and trigger site exposure of the G-tetramer. Furthermore, we identified two pockets within the apical domain of F as possible targets for the trigger sites of G. However, the soluble G-tetramer or the head domain was unable to recognize this epitope (Supplementary Fig. 9f), suggesting that the activation of F may require both the dynamic changes of G-tetramer and the stalk support. In addition, the key G-F interaction sites are conserved only in HNVs, which may be one of the factors that cause HNVs to differ from other species of the henipavirus.

In summary, we isolated a group of protective mAbs against HNV infections with potent neutralizing capacities, diverse epitopes, and multiple mechanisms and are promising drug candidates worthy of further investigation. Furthermore, we described the dynamic structure of the G-tetramer and proposed a new model for G-mediated F activation, providing valuable insights into virus invasion and antibody neutralization and laying a foundation for vaccine design and antibody drug development.

## Methods

### Ethics statement
Rhesus macaque studies were approved by the Institutional Animal Care and Use Committee of the Laboratory Animal Centre of the Academy of Military Medical Sciences (approval no. IACUC-DWZX-2020-052) in compliance with the Laboratory Animal Guidelines for Ethical Review of Animal Welfare (GB/T35892-2018). Hamster studies were performed at the State Key Laboratory of Virology and approved by the Life Science Ethics Committee of the Wuhan Institute of Virology (approval no. WIVA45202105). All animal experiments adhered to the "3R" principle and complied with the relevant provisions of the National Regulations on the Administration of Laboratory Animals.

### Cells and viruses
HEK293T (ATCC, Cat# CRL-11268) cells were cultured at 37 °C and 5% $CO_2$ in Dulbecco's modified Eagle's medium (DMEM) (Gibco, Cat# C11995500BT) supplemented with 10% fetal bovine serum (FBS), 100 μg ml$^{-1}$ streptomycin, and 100 IU/mL penicillin (Gibco, Cat# 15140122). Expi293F human cells (Thermo Fisher Scientific, Cat# A14527) were derived from the 293F cell line and cultivated in Expi293 Expression Medium at 37 °C in a humidified 8% $CO_2$ shaker rotating at 125 rpm. Vero E6 (ATCC, Cat# CRL-1586) cells were maintained at 37 °C and 5% $CO_2$ in DMEM supplemented with 10% FBS. rHIV-HNVs were packaged and applied by the Beijing Institute of Biotechnology. Live $NiV_{MY}$, $NiV_{BD}$, and HeV strains were stored and applied by the Wuhan Institute of Virology.

All procedures involving pseudoviruses were performed under biosafety level-2 conditions, and in vitro and in vivo experiments related to authentic viruses were operated in the biosafety level-4 laboratory.

### Gene construction
Recombinant HNV G constructs and EB2/EB3-Fc chimaeras were generated by modifying previous methods[35,54]. The HNV G constructs for immunization and binding analysis contained the tPA signal peptide, a Strep-tag (WSHPQFEK), a short linker (GGS), and the ectodomains of $NiV_{BD}$ G (residues 71–602, GenBank: AY988601.1), $NiV_{MY}$ G (residues 71–602, GenBank: FN869553.1), or HeV G (residues 71–604, GenBank: NC_001906.3). The construct of $NiV_{BD}$ $G_{HD}$ contained the tPA signal

peptide, a 6×His tag, a short flexible linker (GSGGGS), and the head domain of G (residues 176–602). The construct of $NiV_{MY}$ G used for structural determination comprised the tPA signal peptide, a 6×His tag, and the ectodomain of $NiV_{MY}$ G (residues 72–602, GenBank: NP_112027.1). The ephrinB3-Fc chimaera consisted of an N-terminal tPA signal peptide, residues 28–224 of human ephrin-B3 (GenBank: NM_001406.4), an IEGRMD linker, and residues 100–330 of the human $IgG_1$ heavy chain (GenBank: AXN93646.1). The ephrinB2-Fc chimaera consisted of residues 1–229 of human ephrin-B2 (GenBank: NM_004093.4), an IEGRMD linker, and residues 100–330 of the human $IgG_1$. The soluble construct of NiV F contained the tPA signal peptide, residues 26–482 of the $NiV_{BD}$ fusion protein, the GCNt motif (MKQIEDKIEEILSKIYHIENEIARIKKLIGE), a short linker (GGS), and a 6×His tag. These constructs were codon-optimized, synthesized, and cloned into the pcDNA3.1 (+) expression DNA plasmid.

Full-length HNV G/F ($NiV_{BD}$, GenBank: AY988601.1; $NiV_{MY}$, GenBank: FN869553.1; HeV, GenBank: NC_001906.3; HeV-g2, GenBank: MZ229748.1) and EB2 used for pseudovirus packaging or flow cytometry analysis were synthesized and cloned into pcDNA3.1 vector. A truncated variant with a 22 amino acid deletion at the C-terminal cytoplasmic tail of the henipavirus fusion glycoprotein, T5F, was used to enhance the efficiency of pseudovirus packaging[55]. Truncated forms of HNV F and mutants of HNV G or 1E5 were obtained using the Q5 Site-Directed Mutagenesis Kit (New England Biolabs, Cat# E0552S).

The $V_H$ and $V_L$ genes of m102.4 (Patent No. US14026142B2), h5B3.1 (Patent No. US15951327B2), HENV-26 (Patent No. WO2021097024), HENV-32 (Patent No. WO2022132710), 4H3[43], and 1H8[43], respectively, were added with the constant region of the human $IgG_1$ heavy/light chain before synthesis. The Fab fragments used for structural analysis were produced according to a previously described transient expression route[40,56]. The heavy Fd chain ($V_H$ domain and $CH_1$ domain) with a C-terminal hexahistidine tag ending at $^{222}$CDKTH$^{226}$-HHHHHH was amplified from the entire heavy chain. The full-length antibodies or Fab constructs were cloned into the pcDNA3.4 vector for expression.

### Protein preparation
Proteins were produced in 30 mL of Expi293F cells by transient transfection using the ExpiFectamine 293 Transfection Kit (Thermo Fisher Scientific, Cat# A14526)[40]. Briefly, a mixture of 30 μg of the expression plasmids of HNVs G, NiV sF, EB2/EB3-Fc, mAbs, or Fab and 80 μL of transfection reagent was prepared following the manufacturer's instructions before being added to a 125 mL shaker flask containing 7.5 ×10$^7$ Expi293F cells. Two enhancers were added to the cells 16 h post-transfection, after which the cells were cultivated for another 4–5 days at 37 °C and 125 rpm in 8% $CO_2$. The supernatants were harvested by centrifugation at 800 × g and 5000 × g at 4 °C for 10 min and subsequently filtered through 0.22 μm syringe filters (PALL, Cat# 4612) or disposable vacuum systems (Biosharp, Cat# BS-500-XT). Recombinant proteins were purified on an ÄKTA Pure 150 purification system (GE Healthcare) using a 5 mL StrepTrap HP column (binding buffer: 100 mM Tris, 150 mM NaCl, and 1 mM EDTA, pH 8.0; elution buffer: 2.5 mM desthiobiotin in binding buffer); a HisTrap Excel column (binding buffer: 50 mM Tris, 150 mM NaCl, 10 mM EDTA, and 10 mM imidazole, pH 8.0; elution buffer: 50 mM Tris, 150 mM NaCl, 10 mM EDTA, and 300 mM imidazole, pH 8.0); or a HiTrap rProtein A column (binding buffer: PBS, pH 7.4; elution buffer: 0.1 M glycine, pH 3.0; neutralization buffer: 1 M Tris-HCl, pH 9.0); and buffer exchanged and stored in either PBS (137 mM NaCl, 2.7 mM KCl, 10 mM $Na_2HPO_4$, and 1.8 mM $KH_2PO_4$, pH 7.4) or TBS (50 mM Tris, 150 mM NaCl, and 10 mM EDTA, pH 8.0). All proteins were analyzed for purity by SDS−PAGE (Supplementary Fig. 2b) or high-performance liquid chromatography, and the concentration was determined using a BCA protein assay kit (Merck Millipore, Cat# 23225) or the UV 280 nm absorption method. The 15 aa long with 11 aa overlapped peptides

spanning the entire NiV$_{BD}$ G was synthesized by GL Biochem (Shanghai) Ltd.

## Rhesus macaque immunization

A female rhesus macaque, aged 5 years, was purchased from the Laboratory Animal Centre of the Academy of Military Medical Sciences and housed in a single cage. The animal was intramuscularly vaccinated with $10^9$ infectious units of rAd5-NiV$_{BD}$ and boosted with recombinant NiV$_{BD}$ G on day 28 and HeV G on day 49 via the same route. A total volume of 1 mL of 0.25 mg of aluminum hydroxide adjuvant (InvivoGen, Cat# vac-alu-50), 150 μg of ODN 1826 VacciGrade (InvivoGen, Cat# tlrl-2006-1), and 100 μg of recombinant NiV G or HeV G was used for each booster vaccination. Venous blood samples were collected at 0, 7, 21, 28, 35, 42, 63, and 77 days after the first vaccination. Sera were separated by centrifugation at $5000 \times g$ at 4 °C for 10 min and stored in aliquots at −80 °C for further analysis. An additional 15 mL of fresh anticoagulated blood was collected for single-cell sorting on day 77.

## NiV$_{BD}$ G-specific memory B-cell isolation and single-cell PCR

PBMCs were isolated following the manufacturer's instructions (Dakewe, Cat# 7511011) and washed twice with PBS before being resuspended in FPBS (PBS + 2% FBS). NiV G-specific memory B-cell sorting was performed following a previously described method for sorting human single memory B cells with modifications[40]. Briefly, the PBMCs were incubated with PerCP mouse anti-human/NHP CD3 (BD Pharmingen, clone SP34-2, Cat# 552851, 10 μL/$5 \times 10^5$ cells), APC mouse anti-human CD19 (Beckman, clone J3-119, Cat# IM2470, 10 μL/$5 \times 10^5$ cells), PE-Cy7 mouse anti-human CD27 (Beckman, clone 1A4CD27, Cat# B49205, 10 μL/$5 \times 10^5$ cells), and PE mouse anti-human IgG (BD Pharmingen, clone G18-145, Cat# 555787, 15 μL/$5 \times 10^5$ cells) according to the manufacturer's instructions. Fluorescein isothiocyanate (FITC) (Sigma-Aldrich, Cat# F4274) -labeled NiV$_{BD}$ G was added to the fluorescence cocktails before light-free incubation at 4 °C for 1 h. The cells were washed twice with FPBS and filtered through a 40 μm cell strainer, followed by single-cell sorting using a MoFlo XDP cell sorter (Beckman Coulter). Single NiV G-specific memory B cells labeled CD3$^-$CD19$^+$CD27$^+$IgG$^+$NiV$_{BD}$ G$^+$ were sorted into 96-well PCR plates containing 20 μL of RNase-free water (Transgen, Cat# GI201-01) and 20 U of RNasin ribonuclease inhibitor (Promega, Cat# N2615). The single-cell RT-PCR was performed using SuperScript™ III First-Strand Synthesis System (Invitrogen, Cat# 18080051) with previously reported primers[39] to recover paired variable region genes of mAbs. Amplified V$_H$ and V$_L$ genes were then assembled into full-length linear cassettes containing human IgG$_1$ constant sequences for rapid screening[57].

## ELISA

Detection of serum cross-reactivity with henipavirus G proteins. Microplates (Corning, Cat# 9018) comprising 96 wells were coated with 1 μg ml$^{-1}$ NiV G or HeV G diluted in 0.1 M carbonate buffer (pH 9.6) overnight at 4 °C. The following day, the plates were washed three times with 300 μL of PBS containing 0.2% Tween-20 (PBST) using a 405 TS washer (BioTek) and blocked with 100 μL of blocking solution (PBST + 2% bovine serum albumin (BSA) (Solarbio, Cat# A8010)). The plates were washed before adding 3-fold serially diluted serum starting at 1:100. After incubation, the plates were washed, and a 1:10000 dilution of HRP-conjugated goat anti-monkey IgG H&L antibody (Abcam, Cat# ab112767) was added. These incubations were performed at 37 °C for 1 h. A final wash was conducted before the addition of 100 μL of TMB substrate (Solarbio, Cat# PR1200) for 6 min at room temperature. The absorbance was read at 450 nm and 630 nm on a SpectraMax 190 (Molecular Devices) after the reaction was terminated by adding 50 μL of stop solution (Solarbio, Cat# C1058), and the difference was calculated to eliminate nonspecific absorption.

Screening for G-specific antibodies. A total of $3 \times 10^4$ HEK293T cells were seeded into each well of 96-well plates 1 day before transfection and cultured in 150 μL of DMEM at 37 °C and 5% CO$_2$. Pairs of linear cassettes of heavy and light chains (0.2 μg each) were diluted in serum-free Opti-MEM (Gibco, Cat# 31985-062) and then mixed with 0.8 μL of Turbofect reagent (Thermo Fisher Scientific, Cat# R0532) to a final volume of 40 μL. After incubating for 15 min at room temperature, the mixture was added to each well. The supernatants were collected 48 h post-transfection, and binding activity to NiV$_{BD}$ G was detected via ELISA, as described above, using an HRP-conjugated goat anti-human IgG Fc antibody (Abcam, Cat# 97225).

Detection of the ability of the antibody to bind to henipavirus G proteins. After blocking, the 96-well microplates were incubated with 3-fold serially diluted purified antibodies starting from 5 μg ml$^{-1}$. Subsequently, the steps outlined above were followed.

For peptide scanning, NiV sF was captured on nickel-coated plates (Pierce, Cat# 15442) and then incubated with 10 μg ml$^{-1}$ peptide (15 aa length with 11 aa overlapped) spanning NiV G at 37 °C for 1 h. The plates were washed, 100 μL of 8 ng ml$^{-1}$ h5B3.1 was added, and bound mAbs were detected as described above.

## Pseudovirus packaging and neutralization assay

For pseudovirus packaging, HEK293T cells were seeded into T75 flasks (Corning, Cat# 430720) and cultured at 37 °C and 5% CO$_2$ to 70–90% confluence for transfection. Lipofectamine 3000 (Invitrogen, Cat# L3000015) was used to package the pseudotyped henipaviruses according to the manufacturer's instructions. A total of 20 μg of plasmid (pNL4-3.Luc. R$^-$E$^-$, pcDNA3.1-HNV G, and pcDNA3.1-HNVs T5F at a mass ratio of 8:1:1) and 40 μL of P3000 reagent were diluted in 1 mL of Opti-MEM. The mixture was mixed with another 1 mL of Opti-MEM containing 30 μL of Lipofectamine 3000 reagent, followed by incubation for 15 min at room temperature before addition to T75 flasks. After transfection for 6 h, the culture medium was replaced with 18 mL of fresh medium. The supernatant was collected at 48 h post-transfection by centrifugation at $800 \times g$ for 5 min. The harvested pseudovirus solution was stored in aliquots at −80 °C after filtering through a 0.45 μm filter.

For the neutralization assay, 50 μL of antibodies in 3-fold serial dilutions starting from 10 μg ml$^{-1}$ were incubated with 50 μL of HIV-pseudotyped NiV or HeV ($2 \times 10^4$ TCID$_{50}$/mL) in 96-well plates at 37 °C for 1 h. A volume of 100 μL of DMEM containing $3 \times 10^4$ HEK293T cells was added to each well. After infection at 37 °C and 5% CO$_2$ for 36 h, the medium was removed, and the cells were lysed with 50 μL of lysis buffer (Promega, Cat# E1531) for 10 min at room temperature. A volume of 20 μL of cell lysate was added to 96-well white assay plates (Costar, Cat# 3922), and the light intensity was immediately read on a Spark microplate reader (Tecan) after 50 μL of luciferase assay reagent (Promega, Cat# E1501) was added to each well. The neutralizing ability of the antibodies was calculated by comparing the light intensity of wells in the presence of antibodies to that of wells in the absence of antibodies.

## Luminex assay

Before the experiments, $5 \times 10^6$ MagPlex microspheres (Diasorin, Cat# MC10020-YY) were coupled with 25 μg of NiV$_{BD}$ G, NiV$_{MY}$ G, or HeV G, respectively, using an xMAP Antibody Coupling Kit (Diasorin, Cat# 40-50016) as previously described[38]. A volume of 10 μL of serially diluted serum or antibodies was mixed with 10 μL of NiV$_{BD/MY}$ G or HeV G microspheres (each for 1500) in 96-well black microplates (Corning, Cat# 3916), followed by the addition of 10 μL of biotinylated ephrin B2-Fc or ephrin B3-Fc protein at a final concentration of 45 ng ml$^{-1}$. After incubating at room temperature for 1 h on a plate shaker at 600 rpm, 10 μL of streptavidin-R-phycoerythrin (Invitrogen, Cat# 21627) at 24 μg ml$^{-1}$ was added to the plates. The plates were incubated at room temperature for 30 min on a plate shaker at 600 rpm and then placed

on a magnetic plate separator (Diasorin, Cat# CN-0269-01) to bring all the MagPlex microspheres to the bottom of each well. The plates were washed three times with 100 μL of PBS containing 1% (w/v) BSA and the mean fluorescence intensity was measured using the xMAP INTELLI-FLEX® System (Diasorin).

## Sequence analysis
The sequence details of the variable regions of the IgG heavy and light chains were analyzed using IMGT/V-QUEST (https://www.imgt.org/IMGT_vquest) and compared with the rhesus monkey IG set from the IMGT reference directory[58,59]. Multiple sequence alignment was performed to remove repeat antibodies using ClustalW (https://www.genome.jp/tools-bin/clustalw), and phylogenetic trees were built based on MUSCLE alignment and the neighbor-joining method using MEGA version 7.0.14[60]. Circos plots indicating the combinations of V/D/J alleles were generated using the Circlize v0.4.12 package in R v3.6.3. The vector NTI Advance 11.5.1 (InforMax) was used for general sequence analysis and alignment.

## SDS–PAGE
For SDS–PAGE, 5 μg of reduced or non-reduced protein was loaded onto a SurePAGE™ Plus, Bis-Tris, 4–12% gel (GenScript, Cat# M41210C) and subsequently migrated in Tris-MOPS running buffer at 150 V for 50 min. The gels were stained using an eStain L1 protein staining instrument/kit (GenScript). Images were acquired using an iBright FL1500 imaging system (Invitrogen) in protein gel mode.

## Biolayer interferometry
Binding kinetics and antibody cross-competition were determined by biolayer interferometry assays using the Gator Bioanalysis System (Gator Bio). Antibodies and antigens were diluted in assay buffer (1× PBS, 0.01% BSA, and 0.01% Tween-20) and added to 200 μL of solution in 96-well black flat bottom plates (Greiner Bio-one, Cat# 655209). All assays were performed at 30 °C with agitation at 1000 rpm. A probe loaded with only the buffer served as the baseline for each test column (eight probes).

$NiV_{BD}$ G, used for the binding competition assay, was biotinylated with EZ-Link Sulfo-NHS-LC-Biotin (Thermo Fisher Scientific, Cat# 21335). Gator™ streptavidin probes (Gator Bio, Cat# 160002) were equilibrated for 180 s in assay buffer before being loaded with 20 nM biotinylated $NiV_{BD}$ G for 120 s. Following brief equilibration for 60 s, a primary antibody and nine competing antibodies (including the primary antibody itself) at 100 nM were sequentially captured onto the probes for 300 s. A reference probe loaded with biotinylated $NiV_{BD}$ G, but associated with neither the first nor the second antibody, was used as the background for each test to remove system drift. The percent binding of the competing antibody was calculated as follows: (maximum response in the presence of the primary antibody/maximum response of the competing antibody loaded alone) ×100%.

Binding kinetic analysis was performed as described previously with slight modifications[61]. After pre-equilibration in the assay buffer for 240 s, Gator™ Anti-Human Fc probes (Gator Bio, Cat# 160003) were loaded with antibodies at 1 μg ml⁻¹ for 300 s, followed by the establishment of a 120 s baseline. Subsequently, NiV G or HeV G at various concentrations (800–12.5 nM) were loaded onto the probes for a 300 s association. The dissociation step was performed in assay buffer for 600 s. A reference probe loaded with an antibody without the G protein was used as background for each test to remove system drift.

For the receptor inhibition assay, Gator™ streptavidin probes were loaded with 20 nM biotinylated $NiV_{BD}$ G. The detection antibody used in the competition assay described above was replaced with 100 nM ephrin B2-Fc. The percent inhibition of antibodies was calculated as follows: (1-maximum response of receptor in the presence of antibody/maximum response of receptor in the absence of antibody) × 100%.

For the determination of the affinity of P-59 to sF, biotinylated NiV sF was captured onto Gator™ streptavidin probes. P-59 was dissolved in PBS and diluted to five concentrations at a 2-fold dilution starting from 2 μM, after which the proteins were loaded onto probes for 180 s and dissociated for 900 s.

## Live virus neutralization test
The antibodies were serially diluted in 150 μl of DMEM supplemented with 2.5% FBS and mixed with an equal volume of the virus suspension (1200 PFU/ml). After incubation at 37 °C for 1 h, 250 μl of the mixture (150 PFU/well) was added to monolayer Vero E6 cells in 24-well plates and incubated for another 1 h. Following the removal of the mixture, 0.5 ml of DMEM supplemented with 2.5% FBS and 0.9% methylcellulose (Sigma-Aldrich, Cat# M0555) was added to each well. The plates were incubated in a 5% CO₂-air incubator at 37 °C for 4–5 days. The neutralizing titre was calculated as the reciprocal of the highest antiserum dilution that suppressed plaque formation by 50%. The plaque reduction neutralization titre (PRNT) was calculated as the "inhibitor vs. normalized response (variable slope)" model in GraphPad Prism 8.0 software.

## Cell imaging
Plasmids expressing full-length NiV G (or its mutants) and T5F (or its mutants) were co-transfected into 293T cells in 24-well plates at a ratio of 1:1 (0.25 μg for each). Six hours after transfection, the medium was replaced with 1 ml of fresh medium (for identification of trigger sites) or medium containing 5 μg ml⁻¹ EB2-Fc or antibodies (for membrane fusion inhibition experiment) and incubated for another 24 h. The cells were carefully washed with PBS and then incubated with 500 μl of 0.5× CellMask™ Deep Red plasma membrane stain (Molecular Probes, Cat# C10046) at room temperature for 5 min. Images were acquired using an Operetta CLS High Content Analysis System (PerkinElmer) at 633 nm with a 10× air objective. Representative images (1080 × 1080 pixels) of the same position in different experimental wells were analyzed using ImageJ software (https://imagej.nih.gov/ij/). Unstained black areas and cells larger than 200 pixels were defined as fused parts. The fusion efficiency was calculated as the percentage of the fusion area relative to the total area.

## Flow cytometric analysis
Binding of antibodies to cell surface-anchored G proteins. HEK293T cells grown in T75 flasks were transfected with 15 μg of the pcDNA3.1 plasmid harboring the full-length $NiV_{BD}$ G protein sequence, as described above. After incubating at 37 °C and 5% CO₂ for 36 h, the medium was removed, and the cells were digested with EDTA solution (PBS containing 0.02% (w/v) EDTA). The cells were harvested and washed twice with PBS by centrifugation at 500 × $g$ for 6 min. The cells were resuspended in FPBS and filtered through a 40 μm cell strainer. Approximately 5 × 10⁵ cells were assigned to each tube and incubated with 5 μg ml⁻¹ antibody in 200 μL of FPBS at room temperature for 1 h. The cells were washed twice with 3 mL of FPBS and incubated with 10 μL of PE-conjugated mouse anti-human IgG at room temperature for 1 h. After the final wash, the cells were resuspended in 200 μL of FPBS and loaded onto a FACSCanto II flow cytometer (BD Biosciences). A total of 50,000 events in each tube were recorded and analyzed using FlowJo V10 software. An Ebola virus (EBOV)-specific isotype, mAb 2G1[40], was used as a control.

For the receptor inhibition assay using recombinant proteins, 18 μg of plasmids encoding full-length NiV G, HeV G, or EB2 was transfected into 293 T cells cultured in T75 flasks. The cells were collected 30 h post-transfection and washed twice by centrifugation at 500 × $g$ for 6 min. Approximately 5 × 10⁵ cells were assigned to each tube and then incubated with a mixture of serially diluted antibodies

and 2 µg ml⁻¹ FITC-labeled recombinant EB2-Fc or NiV$_{BD}$ G$_{HD}$ at 4 °C for 1 h. The cells were washed twice and then analyzed on a flow cytometer. In each tube, 10,000 cells were counted. The relative binding percentage was calculated as the ratio of positive cells in the presence or absence of the antibodies. EB2-Fc and G$_{HD}$ served as positive controls, and 2G1 was used as a negative control.

For the receptor inhibition assay using pseudovirions, virus particles were pre-incubated with 5 µg ml⁻¹ of each mAb at 37 °C for 1 h and then added to cells transiently displaying EB2. After incubating for 30 min at 37 °C, the cells were washed, followed by fixation with 200 µl of 4% paraformaldehyde at room temperature for 15 min. The cells were washed and then incubated with 2 µg ml⁻¹ FITC labeled mAb h5B3.1 at 37 °C for 30 min. After the final wash, the cells were analyzed as described above.

For trigger/activation site analysis, 0.25 µg of the wild-type or mutant G/T5F protein full-length expression plasmid was transfected into 24-well plates synchronously with the fusion experiment. Cells were collected 30 h after transfection and detected using 5 µg ml⁻¹ FITC-labeled EB2-Fc or FITC/AF647-labeled G/F-specific mAbs.

### Hamster challenge studies with NiV

Six-week-old female Syrian golden hamsters weighing 100 g were purchased from the Wuhan Institute of Biological Products. After one week of acclimatization, the animals were randomly divided into groups ($n = 6$). All animals were intraperitoneally (i.p.) inoculated with 1000 ×LD$_{50}$ NiV$_{MY}$ in 0.2 mL of PBS on day 0. Animals were given 450 µg or 1 mg of antibody in 0.5 mL of PBS or an equal volume of PBS without antibody via the i.p. route one day before or after infection. All animals were monitored for signs of disease, weighed daily for 14 days post-challenge, and observed daily for an additional 14 days. The moribund and surviving mice were humanely euthanized according to ethical guidelines.

### Crystallization, data collection and processing of the NiV$_{BD}$ G$_{HD}$-1E5 Fab complex

NiV$_{BD}$ G$_{HD}$ was treated with PNGase F (New England Biolabs, Cat# P0704S) at 37 °C for 12 h to release the N-linked glycans. Partially deglycosylated G$_{HD}$ was then incubated with 1E5 Fab at a 1:1.5 molar ratio for 1 h. The G$_{HD}$/Fab complex was purified using a Superdex increase10/300 column with buffer containing 150 mM NaCl, 20 mM Tris, and 1 mM TCEP (pH 8.0) and concentrated to 18 mg ml⁻¹.

Crystallization screening was performed using sitting drop vapor diffusion at 16 °C. Diffraction-quality crystals were obtained from the presence of 1.5 M ammonium sulfate and 0.1 M sodium acetate trihydrate (pH 4.6). Crystals were flash-cooled in liquid nitrogen after cryoprotection in 25% glycerin. The data were collected on the 18U1 beamline at the Shanghai Synchrotron Radiation Facility (SSRF) and processed using HKL2000[62]. Crystal structures were solved using molecular replacement with Phenix[63], using unbound NiV-G (PDB ID: 2VWD) as the search model. Structure refinement was performed using Phenix. The program COOT[64] was used for manual rebuilding, and molecular graphics images were generated using PyMOL (DeLano Scientific LLC, San Carlos, CA). The statistics of the data collection and structure refinement are summarized in Supplementary Table 2.

### Electron microscopy, image processing, and 3D reconstruction

The purified NiV G ectodomain was incubated with an excess molar ratio of purified 1E5 at 4 °C overnight. The complex was purified using Superose 6 Increase 10/300 GL (GE Healthcare, Cat# 29091596) gel filtration in a buffer containing 50 mM Tris/HCl, 150 mM NaCl, and 2% glycerol (pH 8.0). A holey carbon R2/1 200-mesh copper grid with a thin continuous carbon layer (Quantifoil) freshly glowed (10 s at 20 mA) was treated with 0.1% polylysine and air-dried, and 3 µl of 0.16 mg ml⁻¹ NiV G/1E5-Fab complex was loaded onto the freezing mixture using a vitrobot MarkIV (Thermo Fisher Scientific) using a blot force of −1 and 1.5 s at 100% humidity and 4 °C. Sample quality was examined using a Glacios microscope (Thermo Fisher Scientific) operating at 200 kV.

The data were collected using a Titan Krios G3i operated at 300 kV (Thermo Fisher Scientific) instrument equipped with a Gatan K3 detector and a Gif Quantum energy filter with a 20 eV slit width. A total of 19,575 micrographs were automatically acquired with EPU software at a magnification of 105,000× with a pixel size of 0.82 Å and a defocus range of −1.6 to −3.2 µm. Each image comprised 30 frames with an exposure time of 3 s and a dose rate of 17 electrons/second per Å². The data were processed using CryoSPARC3.2.0. After motion correction and estimation of the contrast transfer function, manually picked particles produced a reference for the template picker and were extracted and subjected to iterative 2D classification to remove particles associated with noisy and contaminated classes. In total, 3,851,819 selected particles were extracted to generate an initial model using ab initio reconstruction with no symmetry function in CryoSPARC. A subset of 650,668 particles was selected from these 3D classes for 3D refinement.

The sequence of NiV$_{MY}$ G (72–602 aa) was imported into the SWISS-MODEL server[65] for initial model generation based on PDB 7TY0[35], resulting in a monomer. The initial model of the Fab fragment of the 1E5 antibody was generated using the Alphafold2 multimer[66]. These initial models were combined into an atomic model of the protomer, which was then rigidly fitted to the cryo-EM map, followed by molecular dynamics flexible fitting (MDFF)[67]. The resulting model was optimized using Coot[68] and refined using *phenix.real_space_refine*[63]. The resultant atomic model of the protomer was then fitted to the cryodensity of the other protomer in the cryo-EM map using Chimera[69], followed by optimization using *phenix.real_space_refine*. The final model was evaluated using MolProbity[70]. The statistics of the map reconstruction and model building are summarized in Supplementary Table 3.

### Prediction of G-F interactions based on Discovery Studio

The G$_{HD}$ (PDB ID: 2VSM) and sF (PDB ID: 6TYS) proteins were docked using the Dock Proteins protocol (ZDOCK) in Discovery Studio 4.5. The P-59 peptide of the G protein was restricted from contacting with F, and a total of 54,000 poses of G-F complexes were generated. All the poses were evaluated with a ZRANK scoring function, and the pose with the highest score was used for further analysis.

### Microscale thermophoresis

NiV sF was stained using the NanoTemper Monolith™ RED-NHS 2nd Generation Kit. Approximately 90 µL of 5 µM sF was incubated with 10 µL of 30 µM dye at room temperature for 30 min. After purification on a gel column, the labeled proteins were checked for labeling efficiency, adsorption, and aggregation. Ten microlitres of 16 different concentrations of each peptide (ranging from 1 mM to 30.5 nM) were prepared in PBST and mixed with 10 µL of sF. The mixtures were loaded into Monolith NT.115 serial capillary (NanoTemper Tech), and microscale thermophoresis (MST) signals were recorded at 25 °C, medium power, and 20% excitation power in a Monolith NT.115 (NanoTemper Tech) and analyzed using MO. Affinity Analysis v2.3 software (NanoTemper Tech).

### Quantification and statistical analysis

In the ELISA, neutralization, and Luminex experiments, EC$_{50}$ or IC$_{50}$ values were calculated by fitting a four-parameter curve using Graph-Pad Prism software (version 8.0). In the kinetic analyses, four or five representative curves were fitted globally to a 1:1 Langmuir binding model using Gator™ Part 11 Software (Gator Bio) to calculate the binding kinetics (kon and koff) and affinity values (K$_D$). Dunnett's multiple comparisons test was used to calculate mean differences between experimental and control groups with 95% confidence intervals in flow cytometric and peptide scanning analyses. The log-rank

(Mantel-Cox) test was performed for survival analyses. All the statistical analyses were performed using GraphPad Prism software.

## Reporting summary

Further information on research design is available in the Nature Portfolio Reporting Summary linked to this article.

## Data availability

The crystal structure of 1E5-Fab complexed with the NiV$_{BD}$ G$_{HD}$ and the cryo-EM structures of 1E5-Fab complexed with the NiV$_{MY}$ G-tetramer have been deposited in the Protein Data Bank and Electron Microscopy Data Bank (EMDB) with the accession codes PDB ID 8XC4, PDB ID 8K0C, PDB ID 8K0D, EMDB-36760, and EMDB-36761. This study also used 7TXZ, 7TY0, 2VWD, 2VSM, 2VSK, 3D11, 3D12, 6CMI, 6TYS, and 6VY6 from the Protein Data Bank. Further requests for materials should be addressed to S.C. (qiux@ustc.edu.cn). Source data are provided with this paper.

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

## Acknowledgements

We thank Yi Chen and Feifei Zhao for protein expression and the members of the Laboratory of Advanced Biotechnology for laboratory support. The content of this paper is merely the responsibility of the authors and does not necessarily represent the official views of BIB. We thank Shilong Fan and Min Li from the Technology Center for Protein Sciences, Tsinghua University, for their assistance in crystallography studies. We thank the Cryo-EM Center at the University of Science and Technology of China for supporting cryo-EM data collection. We thank the National Biosafety Laboratory, Wuhan; the Center for Biosafety Mega-Science, Chinese Academy of Sciences; and the National Virus Resource Center for resource support. We thank professor Changming Yu for providing the two anti-F antibodies used in the research. This work was supported by the Defense Industrial Technology Development Program (Grant No. JCKY2020802B001, P.F. and Y.H.L.), the Ministry of Science and Technology of China (Grant No. 2022YFC2303700, K.Z. and S.L.; Grant No. 2022YFA1302700, K.Z.), the Strategic Priority Research Program of the Chinese Academy of Sciences (Grant No. XDB0490000, S.C., R.G., K.Z., and E.L.), the Center for Advanced Interdisciplinary Science and Biomedicine of IHM (Grant No. QYPY20220019, K.Z.), the Fundamental Research Funds for the Central Universities (Grant No. WK9100000032, S.L.; Grant No. WK9100000044, K.Z.), and funding from Hubei Jiangxia Laboratory (Grant No. JXBS002, R.G.).

## Author contributions

S.C., C.Y., K.Z., R.G., and P.F. designed and directed the research; Y.J.L., X.C., G.Z., Y.H.L., Z.S.C., and J.L. immunized the monkey, screened the mAbs, and analyzed sequences; T.F., B.S., Y.R., and Z.L. constructed the plasmids and produced mAbs; L.C. performed the alanine scanning mutagenesis; P.F. generated the proteins, characterized the biological properties of the mAbs, performed the crystal studies, analyzed the mechanisms of mAb neutralizing and G-F interactions, and wrote the draft manuscript; K.Z., M.S., M.L., E.L, and S.L. conducted the cryo-EM structural studies on G/Fab complexes; X.Z., H.Z., Y.Y., C.P., Z.C., and W.G. performed the authentic virus neutralization tests and hamster challenge studies; P.F., S.C., C.Y., K.Z., R.G., M.S., and S.L. edited and revised the manuscript; all authors read and approved the paper.

## Competing interests

C.Y., Y.J.L., P.F., G.Z., Y.H.L., J.L., X.C., and T.F. are inventors on two CN patent applications entitled "Broadly Neutralizing Antibodies Against Henipavirus and Application Thereof" (Patent No. ZL202010713274.0) and "Neutralizing Antibodies Against Nipah Virus and Application Thereof" (Patent No. ZL202010711672.9). These two patents describe antibodies reported in this manuscript and do not impose any restrictions on the publication of the data. The other authors declare no competing interests.
