## [Peer Review File · Nature Communications]

REVIEWERS' COMMENTS

Reviewer #1 (Remarks to the Author):

Thanks for the efforts the author put to revise this article. The reviewer appreciates that most raised concerns have been answered by the author. All antibodies characterization is satisfactory, and G-F interaction identification looks solid. However, there's still some questions and suggestions about G-1E5 structure.

Major:

Section "Dynamic structures of the NiV G-tetramer induced by receptor-like 1E5 Fab"

The reviewer highly suggests removing the comment on the dissociate density labelled as "Head D" in Fig 5a as well as "Head C", as both densities do not have the same size, details and protein like features as the head domain.

Same suggestions on Fig b "Head A/B", especially the neck region looks not resolved.

Line 298-300 "In the loose.....": Please remove this comment as such conclusion can only be made if the stalk region has been fully resolved and the bottom heads are missing.

Line 302-303 "..., which appears...": Please remove this comment as such conclusion can only be made if the neck region and at least one topper head have been fully resolved. If C2 symmetry has been used for author state "compact type", currently "Head A/B" density is highly likely to be artifacts from symmetry application.

Fig 5f: Current representation is a little confusing. It would be appreciated if the author could make the structure difference more visible. Especially the topper heads are solved, so the overlay density looks unnecessary.

Line 322-326 "Compared to", Fig5g: From current view of the map, the neck region in author state "compact type" is not well resolved. Please only make current comments if at least C α tracing is reliable in the neck region.

Line 457-463: Conclusion should be made by evidence rather than assumption. Currently there's no evidence that shows 1E5 binding is exchangeable with EB2, with Fig4 representation being not clear enough and Fig3d showing significant difference in receptor inhibition between 1E5 and EB2-Fc.

Minor:

Line 433-435 "Nevertheless, previous.....": The statement is a little controversial as author cited nAH1.3-G structure is G-tetramer in complex with antibodies.

Reviewer #2 (Remarks to the Author):

There is a tremendous amount of work done and described in the proposed manuscript. There is an incremental improvement, mainly in the details and in the presentation of the results.

However, the major question about the novelty of 1E5 remains unanswered. Even the authors wrote that "we are unable to claim that the identified antibody is superior to m102.4 based on the current data."

Concerning the comparison with the existing antibodies -- the results shown in Fig.2o indicate that 1E5 is not superior to 102.4 no to HENV-26.

Concerning the in vivo protection -- the limitations the team has with the in vivo studies shouldn't serve as an excuse to publish in Nature. The fact is that there is no a proper in vivo efficacy comparison between 1E5 and m102.4.

The answer to my comment 5 is not satisfactory -- although resistant to m102.4 variants were created in vitro, there is no yet Hendra or Nipah isolate not susceptible to 102.4. The examples with Ebola and Covid-19 outbreaks are rather theoretical as I am not aware of any anti-Ebola antibody used in the field and the available anti-covid antibody seems was given only to the president Trump. M102.4 was prophylactically infused in about a dozen individuals. A related article was published in Lancet a few years ago (about the Phase 1 clinical trials in Australia).

REVIEWER COMMENTS

Reviewer #1 (Remarks to the Author):

Thanks for the efforts the author put to revise this article. The reviewer appreciates that most raised concerns have been answered by the author. All antibodies characterization is satisfactory, and G-F interaction identification looks solid. However, there's still some questions and suggestions about G-1E5 structure.

1. Section "Dynamic structures of the NiV G-tetramer induced by receptor-like 1E5 Fab". The reviewer highly suggests removing the comment on the dissociate density labelled as "Head D" in Fig 5a as well as "Head C", as both densities do not have the same size, details and protein like features as the head domain. Same suggestions on Fig b "Head A/B", especially the neck region looks not resolved.
2. Line 298-300 "In the loose.....": Please remove this comment as such conclusion can only be made if the stalk region has been fully resolved and the bottom heads are missing.
3. Line 302-303 "..., which appears...": Please remove this comment as such conclusion can only be made if the neck region and at least one topper head have been fully resolved. If C2 symmetry has been used for author state "compact type", currently "Head A/B" density is highly likely to be artifacts from symmetry application.
4. Fig 5f: Current representation is a little confusing. It would be appreciated if the author could make the structure difference more visible. Especially the topper heads are solved, so the overlay density looks unnecessary.
5. Line 322-326 "Compared to", Fig5g: From current view of the map, the neck region in author state "compact type" is not well resolved. Please only make current comments if at least C α tracing is reliable in the neck region.
6. Line 457-463: Conclusion should be made by evidence rather than assumption. Currently there's no evidence that shows 1E5 binding is exchangeable with EB2, with

Fig4 representation being not clear enough and Fig3d showing significant difference in receptor inhibition between 1E5 and EB2-Fc.

7. Line 433-435 “Nevertheless, previous.....”: The statement is a little controversial as author cited nAH1.3-G structure is G-tetramer in complex with antibodies.

Reviewer #2 (Remarks to the Author):

There is a tremendous amount of work done and described in the proposed manuscript. There is an incremental improvement, mainly in the details and in the presentation of the results.

1. However, the major question about the novelty of 1E5 remains unanswered. Even the authors wrote that "we are unable to claim that the identified antibody is superior to m102.4 based on the current data."

Concerning the comparison with the existing antibodies -- the results shown in Fig.2o indicate that 1E5 is not superior to 102.4 no to HENV-26.

Concerning the in vivo protection -- the limitations the team has with the in vivo studies shouldn't serve as an excuse to publish in Nature. The fact is that there is no a proper in vivo efficacy comparison between 1E5 and m102.4.

2. The answer to my comment 5 is not satisfactory -- although resistant to m102.4 variants were created in vitro, there is no yet Hendra or Nipah isolate not susceptible to 102.4. The examples with Ebola and Covid-19 outbreaks are rather theoretical as I am not aware of any anti-Ebola antibody used in the field and the available anti-covid antibody seems was given only to the president Trump. M102.4 was prophylactically infused in about a dozen individuals. A related article was published in Lancet a few years ago (about the Phase 1 clinical trials in Australia).

Point-by-Point Response

Thank the reviewers for the perspicacious comments and suggestions on our manuscript "A potent Henipavirus cross-neutralizing antibody reveals a dynamic fusion-triggering pattern of the G-tetramer" (NCOMMS-24-14456-T), which are very helpful for us to improve our paper.

We have revised the manuscript in response to the reviewers' comments and provided a detailed point-by-point response as follows (for your convenience, we repeat the referee's comments below in black, followed by our replies in blue).

Reply to Reviewer #1

Thanks for the efforts the author put to revise this article. The reviewer appreciates that most raised concerns have been answered by the author. All antibodies characterization is satisfactory, and G-F interaction identification looks solid. However, there's still some questions and suggestions about G-1E5 structure.

Reply: We are pleased that most of the previous responses and revisions are satisfactory. Based on the reviewer's additional insightful comments, we have made revisions to further improve the manuscript.

1. Section "Dynamic structures of the NiV G-tetramer induced by receptor-like 1E5 Fab". The reviewer highly suggests removing the comment on the dissociate density labelled as "Head D" in Fig 5a as well as "Head C", as both densities do not have the same size, details and protein like features as the head domain. Same suggestions on Fig b "Head A/B", especially the neck region looks not resolved.

Reply: Thanks for suggestions. We have removed the unreasonable descriptions related to Figures 5a and 5b.

Considering that the unbound G-tetramer used has a comparable quaternary structure in negative-stain electron microscopy to that previously reported (Fig. S8a) and that some visible images of weakly solved heads exist in the 2D classification under different orientations, we are confident that the gray-labeled densities in 5a and 5b are

the other two heads of G-tetramer. The flowability in the spatial position makes the density of the mobile heads appear to have different sizes, details, and protein-like features from the well-determined heads in the final models.

2. Line 298-300 “In the loose.....”: Please remove this comment as such conclusion can only be made if the stalk region has been fully resolved and the bottom heads are missing.

Reply: According to your suggestions, we have revised the relevant descriptions.

3. Line 302-303 “..., which appears...”: Please remove this comment as such conclusion can only be made if the neck region and at least one topper head have been fully resolved. If C2 symmetry has been used for author state “compact type”, currently “Head A/B” density is highly likely to be artifacts from symmetry application.

Reply: Thank the reviewer’s professional comments. We have removed the description.

4. Fig 5f: Current representation is a little confusing. It would be appreciated if the author could make the structure difference more visible. Especially the topper heads are solved, so the overlay density looks unnecessary.

Reply: As the comment, the previous representation was inappropriate. We have removed the overlaid densities and made the structure difference more visible.

5. Line 322-326 “Compared to”, Fig5g: From current view of the map, the neck region in author state “compact type” is not well resolved. Please only make current comments if at least C α tracing is reliable in the neck region.

Reply: We have deleted the comments.

6. Line 457-463: Conclusion should be made by evidence rather than assumption. Currently there’s no evidence that shows 1E5 binding is exchangeable with EB2, with Fig4 representation being not clear enough and Fig3d showing significant difference in receptor inhibition between 1E5 and EB2-Fc.

Reply: Thanks for the advice.

To make Fig. 4 more straightforward, we set the structural surface display of EB2, 1E5, and m102.3 with a more pronounced transparency gradient. Fig. 3c and 3d show

significant differences in receptor inhibition between 1E5 and EB2-Fc, but as we analyzed, this may be related to their molecular size. Due to the limited space between the upper and lower heads and between the lower heads and membrane (Fig. S9e), larger IgG molecules (Fig. S2b) may have difficulty sealing off the receptor-binding region of downward heads, which was confirmed indirectly by the strong ability of 1E5 to block the G or virions binding to the membrane-displayed EB2 (Fig. 3e-g).

7. Line 433-435 “Nevertheless, previous.....”: The statement is a little controversial as author cited nAH1.3-G structure is G-tetramer in complex with antibodies.

Reply: There is no ambiguity, as "these antibodies" here refer to the RBD-targeting antibodies mentioned in the previous sentence. The antibody nAh1.3 is not an RBD-targeting antibody.

Reviewer #2 (Remarks to the Author):

There is a tremendous amount of work done and described in the proposed manuscript. There is an incremental improvement, mainly in the details and in the presentation of the results.

1. However, the major question about the novelty of 1E5 remains unanswered. Even the authors wrote that "we are unable to claim that the identified antibody is superior to m102.4 based on the current data." Concerning the comparison with the existing antibodies -- the results shown in Fig.2o indicate that 1E5 is not superior to 102.4 no to HENV-26. Concerning the in vivo protection -- the limitations the team has with the in vivo studies shouldn't serve as an excuse to publish in *Nature* (?). The fact is that there is no a proper in vivo efficacy comparison between 1E5 and m102.4.

Reply: Thanks for the comments. We regret that due to the limited resources of the BSL-4 laboratory, we have not been able to further compare the protective efficacy of these antibodies in animals. In another unpublished study by one of the collaborating teams, prophylactic and therapeutic administration of m102.4 provided survival rates of 83.3% and 16.7%, respectively, in hamsters under the same experimental conditions. Even though m102.4 manifested relatively low survivals, since it was not

a parallel experiment, we cannot claim that it is less potent than the antibody in this study. If conditions permit, we will further compare the in vivo efficacy of these antibodies in the future.

2. The answer to my comment 5 is not satisfactory -- although resistant to m102.4 variants were created in vitro, there is no yet Hendra or Nipah isolate not susceptible to 102.4. The examples with Ebola and Covid-19 outbreaks are rather theoretical as I am not aware of any anti-Ebola antibody used in the field and the available anti-covid antibody seems was given only to the president Trump. M102.4 was prophylactically infused in about a dozen individuals. A related article was published in Lancet a few years ago (about the Phase 1 clinical trials in Australia).

Reply: Sorry for the unsatisfactory responses.

During the Ebola epidemic in West Africa (2013-2016), the antibody cocktail ZMapp was used to treat several patients infected with the Ebola virus. After that, four investigational therapies were applied to 681 patients in a randomized controlled trial of Ebola virus disease treatment conducted in 2018-2019 (ClinicalTrials.gov number, NCT03719586). MAb114 and REGN-EB3 were superior to ZMapp and Remdesivir (N Engl J Med. 2019 Dec 12;381(24):2293-2303.), advancing the approval of two antibody drugs, Ebanga (MAb114) and Inmazeb (REGN-EB3).

Over the past few years, several antibody drugs, such as Evusheld, Sotrovimab, and REGN-COV2, have been approved or urgently authorized to prevent and treat COVID-19. However, as the virus is ever-changing, the efficacy and applicability of these drugs have been more or less affected or even withdrawn from authorization (Sotrovimab). Therefore, expanding the limited antibody candidate pool is essential to cope with possible epidemics, virus variations, or emergencies.

We hope these responses are satisfactory and look forward to hearing from you.

Sincerely,

Pengfei Fan, DSc

Laboratory of Advanced Biotechnology, Institute of Biotechnology

Beijing 100071, China

+86 15201524874

fanpengfei93@163.com